# PUG: Photorealistic and Semantically Controllable Synthetic Data for Representation Learning

**Florian Bordes**[1,2,3]    **Shashank Shekhar**[1]    **Mark Ibrahim**[1]    **Diane Bouchacourt**[1]
**Pascal Vincent**[1,2,3]    **Ari S. Morcos**[1]

[1]FAIR, Meta    [2]Mila - Quebec AI Institute    [3]Université de Montréal, DIRO

## Abstract

Synthetic image datasets offer unmatched advantages for designing and evaluating deep neural networks: they make it possible to (i) render as many data samples as needed, (ii) precisely control each scene and yield granular ground truth labels (and captions), (iii) precisely control distribution shifts between training and testing to isolate variables of interest for sound experimentation. Despite such promise, the use of synthetic image data is still limited – and often played down – mainly due to their lack of realism. Most works therefore rely on datasets of real images, which have often been scraped from public images on the internet, and may have issues with regards to privacy, bias, and copyright, while offering little control over how objects precisely appear. In this work, we present a path to democratize the use of *photorealistic* synthetic data: we develop a new generation of interactive environments for representation learning research, that offer both *controllability* and *realism*. We use the Unreal Engine, a powerful game engine well known in the entertainment industry, to produce **PUG (Photorealistic Unreal Graphics)** environments and datasets for representation learning. In this paper, we demonstrate the potential of PUG to enable more rigorous evaluations of vision models. The datasets can be downloaded at `https://pug.metademolab.com/`.

## 1 Introduction

A grand goal of machine learning is to learn representations of data that are useful across many tasks. Essential to measuring and making progress towards this goal is the availability of ample controllable, realistic data for evaluation and training. This is especially true when considering deep neural network models not only in terms of their raw accuracy, but also their robustness and fairness—crucial properties for models deployed in real-world applications. However, collecting such data is challenging, presenting issues with privacy, bias, and copyright. Furthermore, the majority of available image datasets lack fine-grained labels and are challenging to manipulate beyond coarse image augmentations (e.g. with a photograph, it is hard to change the viewpoint or the time of day).

Using synthetic image data where we precisely control all the factors affecting the rendered scene gives easy access to the corresponding rich set of factor labels. This enables evaluating the extent of a trained deep neural network's abilities, most importantly its robustness. Is the network robust to change in pose? Are the predictions similar for different textures? All these questions may be answered systematically by using synthetic data, enabling highly rigorous evaluations of deep neural network models. In addition, training could also benefit from controllable factors[1], by increasing the robustness of models with respect to these factors. They may also be used to monitor training, e.g. tracking which factors a model focuses on or becomes most invariant to, and in which order, as training progresses. This potentially enables better understanding of the training and generalization

---

[1]We define factors here as distinctive attributes that describe the data, such as color or pose of an object.

37th Conference on Neural Information Processing Systems (NeurIPS 2023) Track on Datasets and Benchmarks.

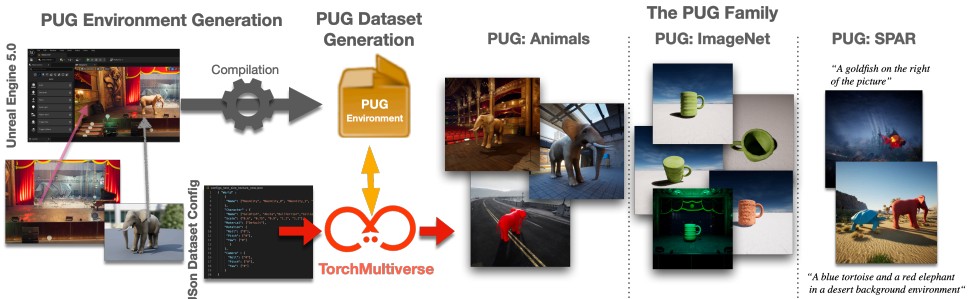

Figure 1: **The PUG Dataset Family** (left) Cartoon illustration of our dataset creation setup, which consists of two steps: environment creation and then data creation. (right) Example images from PUG: Animals, PUG: Image-Net and PUG: SPAR.

dynamics in deep neural networks. However the lack of realism typical in many of the currently available synthetic image datasets, and their usually very limited scope greatly limits their usefulness for general image representation learning research.

To address this, we introduce[2] a new family of synthetic *Photorealistic Unreal Graphics* (PUG) *datasets*, designed for ease of use by the representation learning research community, where image realism is significantly improved compared to current public synthetic image datasets. The environments were built using the Unreal Engine [EpicGames], which is widely used in the video game and entertainment industries and praised for its realism. In addition to pre-rendered static image datasets, we also introduce the TorchMultiverse python library, which offers a simple python interface to enable easily controlled dataset creation from any given PUG environment. Using these tools, we contribute 4 new datasets and show their usefulness across several different research domains. To summarize:

- We introduce a new family of environments and image datasets (coined as PUG) for representation learning, based on the Unreal Engine [EpicGames].

- We present *PUG: Animals* for research on out-of-distribution (OOD) generalization and to study the representational space of foundation models.

- We introduce *PUG: ImageNet* as an additional robustness test set to ImageNet, containing a rich set of factor changes such as pose, background, size, texture, and lighting.

- We introduce *PUG: SPAR* for evaluating vision-language models. We use it to demonstrate how synthetic data can be utilized to address known benchmark limitations. In addition, we introduce *PUG: AR4T* for fine-tuning vision-language models and use it to demonstrate the reliability of PUG: SPAR in contrast to other benchmarks.

## 2   Related work

**Synthetic data for representation learning**   To address robustness shortcomings, researchers today commonly study representations using lower-fidelity controlled datasets such as CLEVR, Biased Cars, and ShapeNet [Johnson et al., 2017, Madan et al., Chang et al., 2015]. Other datasets also contain precise factor labels useful for probing how well a representation encodes each factor in a structured form [Gondal et al., 2019, Struckmeier et al., 2023, Weng, 2018]. While these datasets offer control in terms of the factors that change as well as the train and evaluation splits enabling controlled scientific experimentation, they lack realism. This gap between the lower-fidelity controlled data and the real world poses a challenge for the broader application of these studies. On the other hand, photorealistic datasets have been explored in various application-specific domains in machine learning (outside

---

[2]As a reminder, any use of content or technologies made available by Unreal and/or Epic Games, or any other provider, should comply with their applicable terms (such as the Content License Agreement available at `https://www.unrealengine.com/en-US/eula/content` or any other direct agreement you may have with Epic / Unreal)

of representation learning.) This is especially relevant when trying to evaluate and train models on rare events in which getting real data might be really difficult, such as for autonomous driving. CARLA [Dosovitskiy et al., 2017] is a popular self-driving car simulator which offer highly realistic environment with a significant amount of controllable factors such as environmental conditions, full control of all static and dynamic actors and maps rendering. Another domain where simulated environments are commonly used is reinforcement learning (RL), as RL algorithms often requires the ability to run millions of simulations to learn to master non-trivial tasks, and this cannot be done in a real environment. Data environments based on video games like Atari have been very popular to design and evaluate RL algorithms. Alternatively, platforms like Habitat [Szot et al., 2021] offers indoor scene for training home assistant agents. While these simulators, games or datasets can offer some photo-realism and mimic real world interactions for agents, they are relegated to domain-specific applications making them challenging to use for evaluating the representations of deep neural networks more broadly. Since our focus is not RL, we do not need to embed a fast simulator capable of rendering several thousands frames per second for effective online-training. Instead we can pre-render custom high-quality datasets offline. Photorealistic environments and datasets have also been explored for more general domains with the ThreeDWorld platform [Gan et al., 2021]. Based on the Unity game engine, it offers an interactive environment that can be leveraged to create datasets. The environment is presented as a simulator that is generic enough to handle multiple uses cases, and users can customize the setup of a scene and the data smapling through a low level API. One such dataset that utilizes ThreeDWorld is the Synthetic Visual Concepts (SyVIC) dataset [Cascante-Bonilla et al., 2023], which uses the API to create scene images and descriptive captions for training vision-language models. One of the downsides of ThreeDWorld is that the back-end, the simulator itself, is closed source which limits external contributions. In contrast with ThreeDWorld, we do not provide a platform or a generic simulator for people to use. In fact, we believe that tools like the Unreal Engine are simple enough to be used directly by researchers to create the environments they want without the need to use an intermediate platform. In addition, being free of such intermediate platform allows us to leverage most of the content created for video gaming directly into our simulator by using the existing Epic Games marketplace.

**Evaluating model robustness**   To study model robustness, there is an inherent trade-off between photo-realism and control. Photo-realism depicts objects as they appear in the real world, but often lacks control to precisely define the factors to describe the object such as pose or background. Prior works either collect natural images with specific factor changes [Xiao et al., 2020, Barbu et al., 2019] or label distinctive factors in existing datasets [Idrissi et al., 2022]. Such datasets allow researchers to measure average accuracy on photo-realistic images, but lack granular control necessary for precisely controlled robustness experiments. On the other hand, prior studies [Ibrahim et al., 2022, Abbas and Deny, 2022, Alcorn et al., 2019] examine model robustness with respect to factors such as pose and size by rendering 3D-objects such as buses. These studies precisely control how each object is depicted, but lack in realism. In this work, we advance the photo-realism of these prior works by using the Unreal engine 5.0 [EpicGames], a rendering engine commonly used in high-end cinematic CGI and high-resolution video games which allow us to measure robustness with respect to factors of variation such as lighting.

**Benefits and limitations of using generative models as data generator**   Another way to generate realistic datasets is to use generative models[Ho et al., 2020, Goodfellow et al., 2020]. However, one limitation of such models, despite impressive improvements in the last few years [Dhariwal and Nichol, 2021], is the lack of quality control on what the model can produce [Gandikota et al., 2023]. It's not uncommon to find cases in which the model will ignore parts of the conditioning prompt. Despite such limitation many works have tried to leverage generative model as an additional source of data to train deep neural networks with some success [Astolfi et al., 2023, Bansal and Grover, 2023, Trabucco et al., 2023, Azizi et al., 2023, Zhang et al., 2021, Li et al., 2022a, Jahanian et al., 2022, Jain et al., 2023, Sariyildiz et al., 2023, He et al., 2023]. Another limitation of using generative models are privacy concerns that arise from such models replicating data from their training datasets Somepalli et al. [2022]. Finally, Shumailov et al. [2023] recently demonstrated that training on data recursively generated from such models results in increasing underestimates of the tails and overestimates of the mode, amplifying bias in datasets. In contrast to generative models that might produce unreliable results, we use an entirely controllable environment for which we can have a known and accurate generation with respect to a set of factors.

# 3 Photorealistic Unreal Graphics (PUG) environments and datasets

## 3.1 Leveraging Unreal Engine to create environments and datasets for representation learning

We introduce the *Photorealistic Unreal Graphics* (PUG) environments, a family of 3D graphics environments that leverage Unreal Engine for rendering image data for representation learning research. To create a PUG environment, we first obtain a number of assets [3] which can be 3D objects or 3D backgrounds. Then, we import them in the Unreal Engine editor and create blueprints that yield a simple generic 3D environment. Once this generic and controllable environment is created, it is compiled into a Linux binary file, which can be run on standard GPU clusters. This environment is programmed in such a way that when running, it is listening for incoming packets through WebRTC which can specify instructions about how to change a scene. Since most machine learning practitioners are used to python scripting, we wanted to have a very simple approach by which a user can request image data rendered from a packaged PUG environment, through very simple python code and JSON config files. To do so, we developed a python API, *TorchMultiverse*, that allows a user to easily specify a scene configuration in JSON and request rendered images from the PUG by using WebRTC. Once the factors have been set as requested by the user, for a specific environment configuration, the user can send a command to freeze the current environment and receive back an image. It takes around 1 second to render an image at a resolution of 512x512 on a V100 GPU[4]. We illustrate this setup in Figure 1. It shows how, starting from 3D assets, we design interactive environments that enable us to create different datasets. In the present work, we focus on pre-rendered static image datasets, however our setup also allows dynamic communication between a PUG environment and a pytorch program, meaning that new data could be requested and rendered on the fly while training a deep neural network. We leave the exploration of such active learning setups, as well as the rendering of videos, as future work.[5]

## 3.2 PUG: Animals

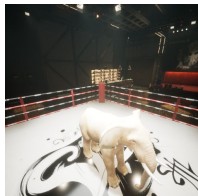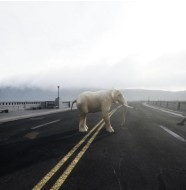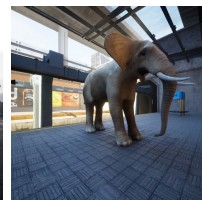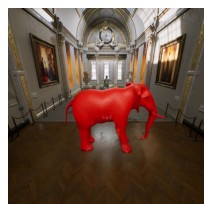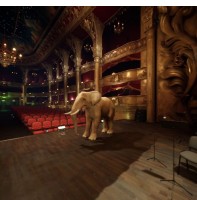

Figure 2: We present *PUG: Animals*, a new photorealistic synthetic dataset with annotated factors of variations to evaluate the out-of-distribution (OOD) robustness of models.

As the first member of the PUG family we introduce *PUG: Animals* (Figure 2), which contains 215 040 pre-rendered images using 70 animals assets, 64 backgrounds, 3 object sizes, 4 textures, under 4 different camera orientations. PUG: Animals is designed with the intent to create a dataset with every combination of the factors of variation available. PUG: Animals allows one to precisely control distribution shifts between training and testing which can give researchers better insight on how a deep neural network generalizes on held out factors of variations. Surprisingly, the usage of 3D realistic synthetic data is limited in OOD generalization research – with the exception of Biased-cars[Madan et al., 2022] that has been used to study generalization on new category-viewpoints. Commons OOD datasets are Colored MNIST [Arjovsky et al., 2020] – to study how well a network can generalize to unseen combinations of digits and colors and MNIST-R [Ghifary et al., 2015] – to study generalization on new combination of digits and rotations. However, MNIST-based dataset

---

[3]We purchased assets from the Epic Game Store and used assets from Sketchfab [Deitke et al., 2022]. The complete list of assets we have used is available at `https://github.com/facebookresearch/PUG`

[4]In our setup, we paralyze the rendering across 64 GPUs. A dataset like PUG: Animals which contains 200K images has taken around 1h to be entirely rendered.

[5]It might also conceivably be used as a photorealistic interactive environment for reinforcement learning (RL), but the high quality image rendering achieved in this system currently appears too compute-intensive and slow to be practically useful in the context of current RL research. Our initial targeted research community and use case is that of supervised and self/unsupervised representation learning from image data, rather than RL.

might be too toyish to evaluate modern architectures. A more *realistic* dataset based on real images is Nico++[Xingxuan Zhang, 2022] – to study generalization with respect to different domains or environments. However, in Nico++ the objects and backgrounds are never entirely disentangle (the context background is different for each image). Thus, it is never clear if the model is failing because of the context or because of a specific object (since the contexts and the objects are never disentangle).

In contrast, in PUG: Animals the animal asset is always the same, in that case the environment factor and the objects are perfectly disentangle such that if the model is able to classify correctly an elephant on a road and is not able to classify the elephant in a museum, we can rigorously say that the failure is caused by the change in context background. In addition of analysis the robustness with respect to the background scene, it is also possible to analyse with PUG: Animal the robustness with respect to the camera position, asset size and texture (as we demonstrated in Appendix 3.2).

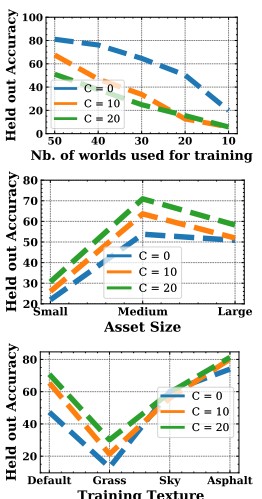

**Classification with held out sets**   In the first experiment, we held out some factors of variation during training (backgrounds, sizes or textures) except for a specific number of animals $C$ and use the held out data as validation set. Thus, $C = 0$ means that the network never saw the factor during training (this is an OOD scenario with unseen factors) while $C = 10$ imply that the network saw this factor for least 10 different animals (OOD scenario with unseen combinations of factors). In Figure 3, we present our results training a ResNet50 with different held out factors. Every model reached more than 99% accuracy on the training set. First, we trained on 50 backgrounds and used the remaining 14 backgrounds for validation: here, the network reached an accuracy of 80%. However, when using only 30 backgrounds for training and using the remaining 34 as validation, the accuracy drop significantly. Interestingly, showing every background for some of the animals (having unseen combination of factors instead of just unseen factors) decrease performance. In contrast, for texture, we found that having at least 10 animals for which every possible textures are seen during training improves generalization. Interestingly, the network overfits much more to the grass texture relative the default network. Lastly, when looking at the size factor, it seems that training on medium size assets leads to good generalization on small and large assets while training only on small asset leads to worse performance on medium and large assets.

Figure 3: Accuracy on held out factors with PUG: Animals. Each line and value C correspond to the number of animals for which all the factors are seen. The test space is built by taking all the factors minus the training factors. If we train on the Default texture, then the network is evaluate on Grass, Sky and Asphalt. If we train on 50 backgrounds, then we evaluate on 64 (total number of background) - 50 (training background) = 14 backgrounds.

**Studying foundation model representational space**   PUG: Animal can also be to study the equivariance of foundation models' representations. For this, we augment each image in PUG: Animals with a caption that describes it according to its factor values (sizes are converted to three adjectives: "small", "medium" or "big", see Appendix C.1 for details), using the following template[6]: *"A photo of a [size] sized [character] textured with [texture] on a [background] background"*. Informally, equivariance of a model's representation with respect to a factor means that when the factor changes from one value to another, the embedding of the corresponding image (or caption) changes in a predictable manner. Equivariance is a sought-after property to improve sample efficiency and robustness to transformations [Klee et al., 2023, Tai et al., 2019, Wang et al., 2023]. Similar to previous works on equivariance and compositionality Bouchacourt et al. [2021], Xie et al. [2022], we measure equivariance as the alignment (i.e. parallelism) between embedding differences. First, we feed images and their corresponding captions to 9 pretrained vision-language models including multiple CLIP models Radford et al. [2021], NegCLIP Yuksekgonul et al. [2023] Flava Singh et al. [2022], BLIP Li et al. [2022b] and X-VLM Zeng et al. [2021] and collect their embeddings of PUG: Animals images and created captions. For each model, we compute difference vectors between the embeddings of two images (or captions) of an object undergoing a factor change: e.g. a big penguin textured with grass on a background "village square" modified to the same penguin but

---

[6]Note that the camera and character orientations are not described, as well as the texture when it is default.

with background "restaurant", see arrows in Figure 4 (left). Specifically, for a sample $i$, undergoing a change from background $b_k$ for background $b_l$, we denote the difference vector between the embedding of the image of the sample with backgorund $b_k$ and the image of the same sample but with background $b_l$ by $v^i_{b_k \to b_l}$. Similarly, we denote by $u^i_{b_k \to b_l}$ the difference vector between the embedding of each of the two captions accompanying the images.

Then, we measure the alignment of difference vectors across pairs *undergoing the same factor change* (here, the penguin and the cat) as their cosine similarity[7]. We estimate three types of equivariance: (i) Image equivariance: how parallel (measured with cosine similarity) are difference vectors across image pairs? (lined and dashed red arrows) (ii) Text equivariance: same but for caption pairs (parallelism of lined and dashed green arrows) (iii) Across modalities equivariance: for the same object, alignment of difference vectors between pairs of image-caption (i.e. alignment of the two arrows for the penguin). Specifically, for image equivariance between sample $i$ and $j$, for background change $b_k$ to $b_l$, we compute:

$$sim(v^i_{b_k \to b_l}, v^j_{b_k \to b_l}) = \frac{{v^i_{b_k \to b_l}}^T v^j_{b_k \to b_l}}{||v^i_{b_k \to b_l}|| \, ||v^j_{b_k \to b_l}||} \tag{1}$$

For text equivariance, we compute be $sim(u^i_{b_k \to b_l}, u^j_{b_k \to b_l})$ while for across equivariance, we compute $sim(v^i_{b_k \to b_l}, u^i_{b_k \to b_l})$. We report cosine similarity averaged over pairs and possible changes for each factor (higher value means higher equivariance, 1 is the maximum). Specifically, the image equivariance for background writes as

$$\frac{1}{B(B-1)} \sum_{b_k} \sum_{b_l} \frac{1}{N(N-1)} \sum_i \sum_j sim(v^i_{b_k \to b_l}, v^j_{b_k \to b_l}) \tag{2}$$

where $B$ is the number of possible backgrounds and $N$ is the number of samples.

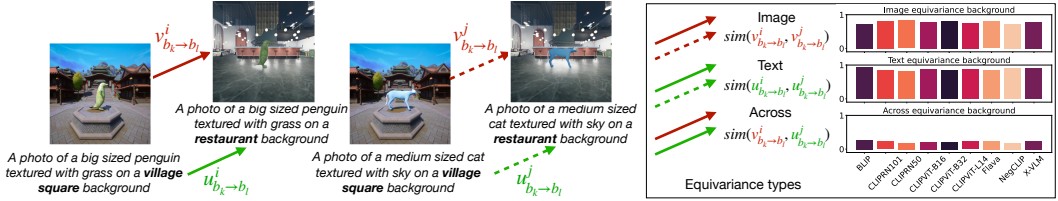

Figure 4: **Measuring foundation models equivariance thanks to PUG: Animals. Left:** Illustration of how to use PUG: Animals to compute equivariance. **Right:** Image and text equivariance is present with respect to background, while across modalities equivariance to background doesn't hold as much. See main text for detailed results.

We show in Figure 4 (right) results for equivariance with respect to the background. Plots for equivariance to texture and size are in Figure 10. Looking at Figure 4 results (right side, top row), we see that the foundation models' image embeddings present high equivariance to background ($0.78 \pm 0.04$ on average over models). There is also (see Figure 10a) small image equivariance to texture ($0.15 \pm 0.04$), but almost no equivariance to size ($0.06 \pm 0.02$). Text equivariance is high with respect to background (average of $0.87 \pm 0.03$), but is also strong for size and texture ($0.71 \pm 0.11$ for size and $0.81 \pm 0.03$ for texture, see Figure 10b) suggesting that foundation models' caption embeddings can be manipulated with vector arithmetic, similar to word vectors behaviours Ethayarajh et al. [2019]. This aligns with the recent work of [Trager et al., 2023] that show linear behavior of VLMs text embedding spaces. Across modalities, small equivariance is present with respect to background ($0.22 \pm 0.03$ and Figure 4 right side, bottom row). However when size or texture change for a given object, its image and caption representations seem to move in non-aligned directions ($0.07 \pm 0.01$ for texture and $0.04 \pm 0.01$ for size, see Figure 10c). While more syntactically complex captions and other equivariance metrics could be designed, our aim here is to provide an initial study to showcase how PUG: Animals can be easily used to study state-of-the-art models representations.

---

[7]Note that foundation model representations belong to the hypersphere, yet measuring equivariance as parallelism relies on Euclidean geometry, we discuss this in Appendix C.1. Still, cosine similarity is a starting point to showcase how PUG: Animals can be used to study models' representations.

## 3.3 PUG: ImageNet

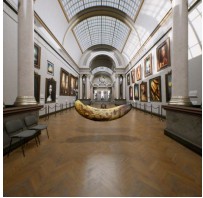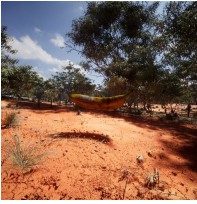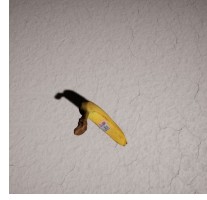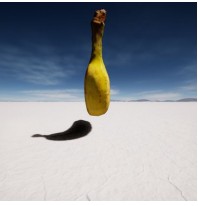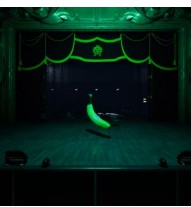

Figure 5: We present **PUG: ImageNet**, a new photorealistic synthetic dataset with annotated factors of variations as an additional test set for ImageNet pretrained models.

As a second member of the PUG family, we introduce *PUG: ImageNet* (Figure 5), which contains 88,328 pre-rendered images using 724 assets representing 151 ImageNet classes with 64 backgrounds, 7 sizes, 9 textures, 18 different camera orientation, 18 different character orientation and 7 light intensity. In contrast to PUG: Animals, PUG: ImageNet was created by varying only a single factor at a time (which explain the lower number of images than PUG: Animals despite using more factors). The main purpose of this dataset is to provide a novel useful benchmark, paralleling ImageNet, but for *fine-grained evaluation of the robustness* of image classifiers, along several factors of variation.

**An extensive evaluation of the robustness of SOTA models**  Our PUG: ImageNet dataset offers both photo-realism and precise control over how each object is depicted from pose and size to environment and camera-angle. We also provide a collection of objects with mappings to classes in the popular ImageNet dataset, enabling researchers to probe the robustness of SoTA vision models without retraining. We assess a variety of model architectures across several pretraining datasets including ImageNet-1/-21k, LAION (400M and 2B), and JFT300M [Kolesnikov et al., 2020, Liu et al., 2021, Dosovitskiy et al., 2021]. We observe in Table 1 that the models that perform the best on the ImageNet validation accuracy are not always the ones which offer the best robustness on PUG: ImageNet. For example, the pretrained ViT-B32 trained on ImageNet-21k is better on the ImageNet validation set compared to a Swin-B, but offers worse robustness across all factors. We confirm no statistically significant relationship exists between ImageNet accuracy and robustness by computing Pearson's correlation coefficients (Appendix C.3). This result showcases how PUG: ImageNet can be added as an additional benchmark to evaluate vision models.

| | | PUG: ImageNet Top-1 Accuracy across Factors of Variation | | | | | |
|---|---|---|---|---|---|---|---|
| | ImageNet Val. | Camera (Yaw,Pitch,Roll) | Pose (Yaw,Pitch,Roll) | Size | Texture | Light | Background |
| ResNet50 | 81.5 | (38.1, 33.1, 26.9) | (38.0, 23.6, 22.9) | 35.7 | 27.0 | 13.6 | 29.5 |
| ResNet101 | 82.3 | (43.4, 35.9, 29.4) | (45.1, 26.7, 25.6) | 39.7 | 31.1 | 14.1 | 32.8 |
| ViTLarge | 85.8 | (52.2, 40.4, 37.1) | (52.4, 30.4, 28.4) | 46.4 | 42.9 | 8.9 | 34.6 |
| ViTBasePretrained21k | 84.3 | (37.5, 34.3, 31.7) | (38.0, 21.8, 20.5) | 33.0 | 28.5 | 4.1 | 26.6 |
| Swin | 83.6 | (56.0, 45.6, 41.8) | (56.9, 35.3, 34.2) | 52.9 | 40.1 | 19.1 | 42.0 |
| BiT (JFT300M) | 80.3 | (40.5, 32.3, 26.0) | (42.1, 23.6, 22.8) | 37.3 | 23.4 | 6.3 | 20.5 |
| DINOv2 (LVD-142M) | 84.5 | (45.6, 41.1, 37.4) | (47.5, 28.8, 28.5) | 43.1 | 35.0 | 6.1 | 30.9 |
| Flava (PMD 70M) | 75.5 | (31.7, 23.4, 17.6) | (30.8, 17.6, 15.4) | 30.5 | 24.2 | 7.8 | 21.9 |
| CLIPViTB32 (400M) | 62.9 | (41.7, 30.2, 22.1) | (41.6, 23.8, 20.9) | 40.1 | 34.4 | 5.7 | 24.4 |
| CLIPViTB32 (2B) | 66.6 | (44.0, 31.5, 24.1) | (43.8, 24.8, 21.8) | 42.2 | 34.7 | 3.3 | 26.0 |
| CLIPViTL14 (400M) | 72.8 | (52.3, 39.8, 35.7) | (51.8, 29.0, 26.4) | 50.6 | 41.1 | 4.3 | 33.0 |

Table 1: Robustness measured by average top-1 accuracy across factors on PUG: ImageNet (We show on the second column the traditional ImageNet validation set accuracy for comparison). Pretraining dataset sizes are indicated in parenthesis with the default being ImageNet-1k. CLIP uses ViT-B32 or ViT-L14. Camera orientation and object pose indicate accuracy along (yaw, pitch, roll) axes.

## 3.4 PUG: SPAR for VLMs

As a third member of the PUG family, we introduce *PUG: SPAR (Scene, Position, Attribute and Relation)* for evaluating vision-language models (VLMs). In contrast to pure vision based models, VLMs should be able to predict the correct caption (from a given set of captions) that describe the content of a given image. Several benchmarks to evaluate VLM models already exist such as Winoground [Thrush et al., 2022] or ARO [Yuksekgonul et al., 2023]. However, recent works[Thrush et al., 2022, Diwan et al., 2022] have highlighted an important limitation in these benchmarks: some image-caption pairs in Winoground might be even too difficult to solve for a human whereas ARO

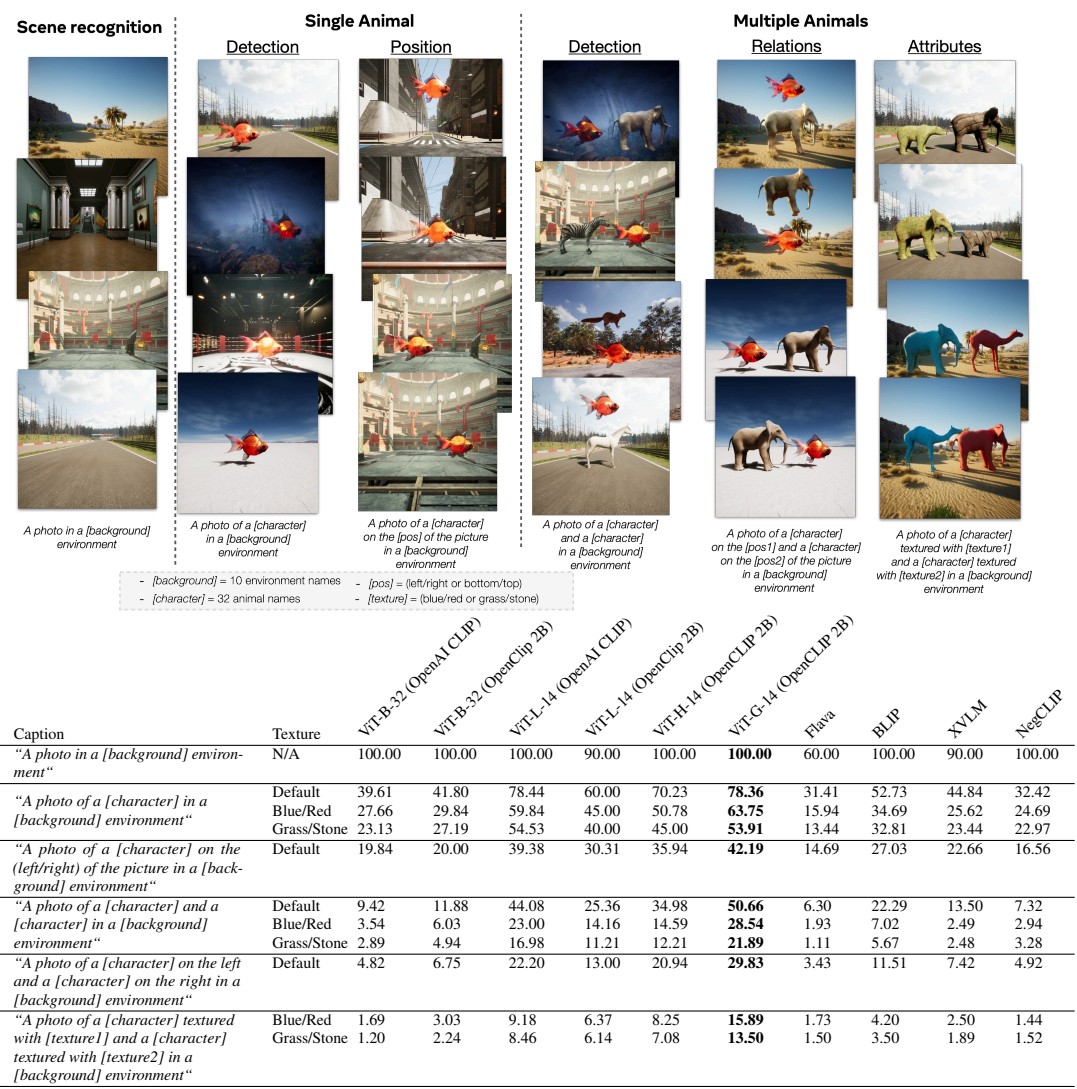

| Caption | Texture | ViT-B-32 (OpenAI CLIP) | ViT-B-32 (OpenClip 2B) | ViT-L-14 (OpenAI CLIP) | ViT-L-14 (OpenClip 2B) | ViT-H-14 (OpenClip 2B) | ViT-G-14 (OpenCLIP 2B) | Flava | BLIP | XVLM | NegCLIP |
|---|---|---|---|---|---|---|---|---|---|---|---|
| *"A photo in a [background] environment"* | N/A | 100.00 | 100.00 | 100.00 | 90.00 | 100.00 | **100.00** | 60.00 | 100.00 | 90.00 | 100.00 |
| *"A photo of a [character] in a [background] environment"* | Default | 39.61 | 41.80 | 78.44 | 60.00 | 70.23 | **78.36** | 31.41 | 52.73 | 44.84 | 32.42 |
| | Blue/Red | 27.66 | 29.84 | 59.84 | 45.00 | 50.78 | **63.75** | 15.94 | 34.69 | 25.62 | 24.69 |
| | Grass/Stone | 23.13 | 27.19 | 54.53 | 40.00 | 45.00 | **53.91** | 13.44 | 32.81 | 23.44 | 22.97 |
| *"A photo of a [character] on the (left/right) of the picture in a [background] environment"* | Default | 19.84 | 20.00 | 39.38 | 30.31 | 35.94 | **42.19** | 14.69 | 27.03 | 22.66 | 16.56 |
| *"A photo of a [character] and a [character] in a [background] environment"* | Default | 9.42 | 11.88 | 44.08 | 25.36 | 34.98 | **50.66** | 6.30 | 22.29 | 13.50 | 7.32 |
| | Blue/Red | 3.54 | 6.03 | 23.00 | 14.16 | 14.59 | **28.54** | 1.93 | 7.02 | 2.49 | 2.94 |
| | Grass/Stone | 2.89 | 4.94 | 16.98 | 11.21 | 12.21 | **21.89** | 1.11 | 5.67 | 2.48 | 3.28 |
| *"A photo of a [character] on the left and a [character] on the right in a [background] environment"* | Default | 4.82 | 6.75 | 22.20 | 13.00 | 20.94 | **29.83** | 3.43 | 11.51 | 7.42 | 4.92 |
| *"A photo of a [character] textured with [texture1] and a [character] textured with [texture2] in a [background] environment"* | Blue/Red | 1.69 | 3.03 | 9.18 | 6.37 | 8.25 | **15.89** | 1.73 | 4.20 | 2.50 | 1.44 |
| | Grass/Stone | 1.20 | 2.24 | 8.46 | 6.14 | 7.08 | **13.50** | 1.50 | 3.50 | 1.89 | 1.52 |

Figure 6: Setup and zero-shot evaluation of CLIP models on PUG: SPAR with **caption retrieval**. By using synthetic data, we can increase progressively the *difficulty* of a scene. Our setup is presented in the image above the table in which we show 6 different types of image captioning. 1) caption for background scene recognition for which we have 10 different backgrounds which are easy to distinguish from each other. 2) caption for single animal class prediction, the model should predict the correct categories over the 32 possible animals and 10 backgrounds (for a total of 320 captions). 3) caption for single animal position prediction that increases the number of caption up to 640 and lead to a significant drop in accuracy for every models. 4) caption for two animals class prediction, the model should predict the correct categories of the two animals presented in the images (5120 captions). 5) caption for two animals positions prediction, the model should predict the position of the two animals in the picture (over a total of 10240 captions). 6) caption for two textured animals class prediction, the model should recognize a blue elephant from a red camel. The performances of several VLMs models are presented in the table for which each row corresponds to one of the scenario described previously.

has been shown by Lin et al. [2023] to be mostly solvable without even using the image information at all. Consider that for an image containing a horse eating grass, ARO will propose two captions: *"the horse eating the grass"* and *"the grass eating the horse"*. The model should predict the correct caption between these two. However, the second caption is impossible, so even without looking at the image, any model can be confident that the first caption is the correct one.

Another shortcoming of current benchmarks is that most of them probe only if the model is correctly able to understand the relations or the attributes between objects. However it is not clear if the failures in finding the correct relations or attributes come from the model not understanding them or come

from not understanding which objects are present in the scene. For example, to understand complex relations like: *A photo of an elephant on the left and a camel on the right in a desert background*, the model should first be able to identify whether the background of the picture is an actual desert. Then, the model should identify whether there is an elephant on the picture. It should understand what is *an elephant on the left*. The model could be very effective at identifying individual elephants or camels, but it could unexpectedly fail when a camel and an elephant appear in the same picture. If the model does not fail in recognizing the animals, then we can probe the model to evaluate the position of each of them. We built PUG: SPAR with this goal of having a progressive evaluation scheme in which we can easily determine exactly what are the failure modes of a given VLM. From basic scene understanding to complex relations and attributes, our dataset offer a simple and yet effective way to get a better understanding of the capabilities and limitations of VLMs.

The dataset contains 43,560 images with the associated factors of variations: 10 backgrounds, 32 animals, 4 relations (left/right, botton/top) and 4 animal texture attributes (blue/red, grass/stone). We have images containing either 1) only the background (for scene recognition) 2) only one animal at different left/right or bottom/top position 3) two animals at different left/right or bottom/top position. And for each of the scenes (either single or multiple animals), we vary the texture of the animals and the background to evaluate the robustness of the model. Our setup and experiments are presented in Figure 6 in which we display some images from the dataset with the corresponding captions used for evaluations. In our benchmark, we used 6 different types of captions to evaluate the following: 1) Scene Recognition (first row) 2) Single animal classification (second row) 3) Single animal position detection (third row) 4) Multiple animals classification (forth row) 5) Multiple animal position detection (fifth row) 6) Multiple animal and textures prediction (sixth row). For each of them, we evaluate the top-1 retrieval accuracy of the correct captions within the set of captions associated to each setup. We evaluate multiple models on these setups: OpenAI CLIP [Radford et al., 2021], OpenCLIP [Ilharco et al., 2021], Flava [Singh et al., 2022], BLIP [Li et al., 2022b], X-VLM [Zeng et al., 2021] and NegCLIP [Yuksekgonul et al., 2023]. Most of the models are correctly able to solve the scene recognition task (which is not surprising since we used only 10 environments which are very different from each other). Concerning the simple object recognition task when using a single animal, the performances across models is highly variable. Our experiments also highlight that the VLM performance in a multiple animals detection setting are much worse than the performance in a single animal detection setting. Those experiments show that despite their successes, VLMs are far from having a good understanding of the world and that improving the robustness of these models is a needed step for real-world robustness.

Inspired by Winoground [Thrush et al., 2022], we present an experimental setup in which we leverage hard-negative pair of images. Instead of performing caption retrieval within all captions associated to a given setup, we performed caption retrieval between the correct and the *hard negative* caption. For example, the hard negative caption of *"An elephant on the left of the picture and a camel on the right of the picture"* will be *"A camel on the left of the picture and an elephant on the right of the picture"*. In addition of switching the relation (left/right and bottom/top), we also provide hard negative captioning for the attributes (blue/red and grass/stone). In Table 2, we present our results using the hard-negative pair. We clearly observe that none of the models are able to predict the correct captions, with many models being close to random performance (50%).

### 3.4.1 PUG: AR4T

Lastly, we introduce PUG: AR4T (Attributes and Relations for training). In contrast to PUG: SPAR which is only an evaluation benchmark, PUG: AR4T was created as an additional fine-tuning dataset for VLMs.[8] As shown in the previous section, VLMs struggle to understand spatial relations or attributes and thus are good candidates for our fine-tuning scenario. PUG: AR4T contains 249,986 training images with captions and 23,216 test images[9]. In Table 3, we present CLIP fine-tuning results on the ARO and PUG: SPAR benchmark. We also compare our results against Syn-CLIP, which is CLIP fine-tuned on the SyVIC synthetic dataset. Our results are very similar to Syn-CLIP, but Syn-CLIP training requires several additional tricks to arrive at this performance (Section 3.2 in

---

[8]The assets used to create PUG: SPAR and PUG: AR4T are different enough such that PUG: SPAR is still a good benchmark to evaluate models fine-tuned with PUG: AR4T.

[9]We also run experiments with a version of this dataset which contain 1M images but as shown in Table 3, adding more images does not increase performance on PUG: SPAR. We only release publicly the version of PUG:AR4T that contain 249,986 images.

| Caption | Values | B-32 CLIP | B-32 OpenClip 2B | L-14 CLIP | L-14 OpenClip 2B | H-14 OpenCLIP 2B | G-14 OpenCLIP 2B | Flava | BLIP | XVLM | NegCLIP |
|---|---|---|---|---|---|---|---|---|---|---|---|
| *"A photo of a [character] on the [position] of the picture in a [background] environment"* | Left/Right | 49.53 | 47.66 | 50.00 | 46.72 | 49.84 | 50.31 | 49.22 | 51.88 | **52.03** | 48.91 |
| | Bottom/Top | 54.84 | 52.34 | **66.87** | 60.62 | 56.56 | 58.91 | 50.47 | 54.84 | 53.28 | 54.06 |
| *"A photo of a [character] on the [position1] and a [character] on the [position2] in a [background] environment"* | Left/Right | 53.88 | 55.62 | 53.17 | **56.23** | 55.74 | 54.44 | 54.41 | 53.79 | 55.02 | 54.36 |
| | Bottom/Top | 51.15 | 53.84 | 54.06 | 53.51 | 57.24 | 56.49 | 55.23 | **60.26** | 58.87 | 54.09 |
| *"A photo of a [character] textured with [texture1] and a [character] textured with [texture2] in a [background] environment"* | Blue/Red | 52.77 | 53.63 | 54.43 | 56.94 | 55.48 | 54.42 | 54.22 | **57.32** | 56.19 | 51.74 |
| | Grass/Stone | 52.79 | 54.14 | 56.31 | 57.28 | 56.62 | **57.19** | 54.53 | 53.94 | 54.26 | 49.92 |

Table 2: We present the performances of several VLMs with **hard negatives captioning** on PUG: SPAR in which we perform retrieval between two captions: the correct caption and the hard-negative corresponding caption. In that instance, the model should choose the correct caption between both of them (the probability to get the correct one with a random model would be 50%). Interestingly, none of the model presented in this table seem to be able to get a real understanding of simple position (left, right, bottom, top) or colors.

[Cascante-Bonilla et al., 2023]). On the other hand, the photo-realistic nature of PUG: AR4T enables us to match Syn-CLIP without any of these additional bells and whistles. However, even if we note some improvements on ARO, we are still far from having a model able to understand spatial relations. This is highlighted by the results given on our PUG: SPAR benchmark for which the improvement on single animal position prediction is still only above random chance while there is no improvement on the double animal location prediction task. This confirm the unreliability of the ARO benchmark highlighted by Lin et al. [2023].

| | VG-Relation (Macro-Accuracy%) | VG-Attribution (Macro-Accuracy%) | ARO COCO-Order (Precision@1) | Flickr30k-Order (Precision@1) | Average | PUG: SPAR (left/right) Single (Precision@1) | Double (Precision@1) |
|---|---|---|---|---|---|---|---|
| **CLIP-ViT-B/32 (400M)** | 59.16 ǀ 55.50 | 62.18 ǀ 61.52 | 47.96 | 59.98 | 57.32 | 49.84 | 54.42 |
| *+ FT w/ Syn-CLIP* | **71.40** | **66.94** | 59.06 | 70.96 | 67.09 (+9.77) | N/A | N/A |
| *+ FT w/ PUG:AR4T (200K)* | 68.36 ǀ 75.18 | 65.54 ǀ 64.44 | 57.80 | 69.74 | 65.36 (+8.04) | 50.78(+0.94) | 54.23(-0.19) |
| *+ FT w/ PUG:AR4T (1M)* | 71.03 ǀ 76.57 | 65.15 ǀ 64.32 | **61.07** | **72.84** | **67.52** (+10.3) | 50.16(+0.32) | 54.19(-0.23) |

Table 3: Fine-tuning CLIP on PUG: AR4T. For VG-Relation and Attribution, the results (Acc1 ǀ Acc2) indicate macro-accuracy across all relations and attributes (Acc1), and macro-accuracy on the subset of relations and attributes present in both ARO and PUG (Acc2). For PUG: SPAR, we evaluate on images in which there is only one animal (Single) or two animals (Double) with the relation being left or right. We were not able to run SynCLIP on PUG: SPAR because the model was not public at the time of the publication.

# 4   Conclusion

The fine-grained controllability of synthetic rendered image data makes it ideal for designing challenging evaluation benchmarks to better understand the properties and limitations of vision models, as well as for controlled training scenarios – if only it was closer to real data. To this effect, we introduced PUG datasets for representation learning. By leveraging the photorealism of the Unreal Engine[EpicGames], we created 4 new datasets and showcased their utility for robust evaluation. We showed how PUG: Animals could be leveraged for OOD generalization and to study properties of the representation spaces. We developed PUG: ImageNet as a new challenging benchmark that researchers can easily use to assess and compare the robustness of image classifiers. With PUG: SPAR we provide a reliable benchmark for vision-language models while PUG:AR4T offer additional data that could be leveraged to fine-tune VLMs. Together, the PUG family of datasets represents a new standard of photorealism and control for synthetic image data.

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

# A   Limitations and Future Work

## A.1   Limitations

In this work, we introduced 4 new datasets that were created using the Unreal Engine. We presented a set of case studies, demonstrating how these datasets can be leveraged to improve both evaluation and training for representation learning. Deeper work would be needed to fully unlock the potential these datasets can offer. In addition, we merely scratch the surface of what a powerful engine such as the Unreal Engine can offer. With advanced techniques such as Lumen, Nanite and Megascans, it is now possible to create even more realistic environments. In addition, the datasets we provide have a single simple label, whereas future uses could easily provide detailed rich labels for the entire scene underlying each generate image.

## A.2   Future Work

A long vision for the PUG family would be to yield a series of benchmarks that can probe the robustness of vision models. PUG: ImageNet is a first step in this direction, however we might want to get more factors of variation such as weather and occlusion. A second take would be to increase the richness of the labelling by making available detailed segmentation masks and labels. Making a short video dataset in which we have complete control over the factors is also a promising future direction since AI research on video is still far behind what can be done with images. The active learning pipeline direction is also worth to explore since the model could bias the PUG environment to produces samples that are the best for a specific downstream task[Mishra et al., 2022].

# B   PUG Datasets

The datasets PUG: Animals, PUG: ImageNet, PUG: SPAR and PUG:AR4T are available under the **cc-by-nc license with the restrictions that they should not be use to train generative AI models**. They are available to download on the following website: `https://pug.metademolab.com/`. The datasets can be read by a torchvision ImageFolder. We have one class by folder and all the images associated to one class are saved in this folder as png. There is a csv file associated to each dataset that map a filename (with unique ID) to its associated factors of variations. Examples of dataloaders are available at `https://github.com/facebookresearch/PUG`.

## B.1   Datasheet

| MOTIVATION | |
|---|---|
| **For what purpose was the dataset created?** | The 4 datasets we presented in this paper were created for representation learning research. PUG: Animals is a strong dataset for OOD research as well as for being able to better probe the representation of vision models. PUG: ImageNet was designed as an additional benchmark for ImageNet pretrained model to offer a better comprehension of vision models capabilities in term of robustness to specific factor of variations. PUG: SPAR showcase how synthetic data can be used to evaluate VLMs understanding while PUG: AR4T can be leveraged to fine-tune them. |
| **Who created the dataset and on behalf of which entity?** | This dataset was created by the FAIR team at Meta AI. |
| **Who funded the creation of the dataset?** | Meta. |
| COMPOSITION | |

| | |
|---|---|
| **What do the instances that comprise the dataset represent?** | The instances represent images of animals in various environment for PUG: Animals. In contrast PUG: ImageNet contains 151 object classes (the full list is available in Appendix B.3). PUG: SPAR uses the sames assets as PUG: Animal while PUG: AR4T use the same objects as PUG: ImageNet. |
| **How many instances are there in total?** | **PUG: Animals**: 215 040 images; see Appendix B.2.

**PUG: ImageNet**: 88,328 images; see Appendix B.3.

**PUG: SPAR**: 43,560 images; see Appendix B.4.

**PUG: AR4T**: 249,986 images for training and 23,216 test images; see Appendix B.5. |
| **Does the dataset contain all possible instances or is it a sample (not necessarily random) of instances from a larger set?** | PUG: Animals contains all possible combination of factors of variations. In contrast PUG: ImageNet was sampled by changing only 1 factor at a time and is therefore a random sample of the distribution. Images in PUG: SPAR were sampled using all possible combination of factors of variations (with the exception that for the attributes the blue or grass animal is always on the left). Image-text pairs in PUG: AR4T were randomly sampled. |
| **What data does each instance consist of?** | For PUG: Animals, PUG: ImageNet, PUG: SPAR we release images along the factor of variation. For PUG: SPAR, we release the script to generate the captions from the factors of variations. For PUG: AR4T, we release images along with corresponding captions. |

| | |
|---|---|
| **Is there a label or target associated with each instance?** | Yes, a csv file. Each instance have a row in this csv files with all the factors of variation used to generate this image. For PUG: Animals, the csv files contains the following columns:

`filename, world_name, character_name, character_scale, camera_yaw, character_texture`

while for PUG: ImageNet, it contains:

`filename, world_name, character_name, character_label, character_rotation_yaw, character_rotation_roll, character_rotation_pitch, character_scale, camera_roll, camera_pitch, camera_yaw, character_texture, scene_light.`

For PUG: SPAR, the csv contains:

`filename, world_name, character_name, character2_name, character1_pos, character2_pos, character_texture, character2_texture`

For PUG: AR4T, the csv contains:

`Relation, Actor1Category, Actor2Category, Actor1Name, Actor2Name, Actor1Location, Actor2Location, Actor1Rotation, Actor2Rotation, Actor1Scale, Actor2Scale, Actor1Texture, Actor2Texture, Actor1Attribute, Actor2Attribute, Camera_roll, Camera_pitch, Camera_yaw, caption, alt_caption, Level, World.Name, filename, filename_neg, filepath` |
| **Is any information missing from individual instances?** | No, all relevant information is included. |
| **Are relationships between individual instances made explicit?** | N/A. |
| **Are there recommended data splits?** | There is no specific split concerning PUG: Animals because this dataset should be used for OOD research. We primarily let the researchers choose their own held out or training/validation/testing split to train their models. In contrast, PUG:ImageNet and PUG:SPAR should only be used as an additional test set. For PUG: AR4T, splits are described in Appendix B.5. |
| **Are there any errors, sources of noise, or redundancies in the dataset?** | We did not explicitly filter for occlusion, so some images may contain occluded views. PUG: Animals and PUG:SPAR are very clean and each animal is easily identifiable. In contrast, PUG: ImageNet and PUG: AR4T leverage assets from Sketchfab and the asset quality vary significantly. |

| | |
|---|---|
| **Is the dataset self-contained, or does it link to or otherwise rely on external resources?** | The dataset is self-contained however the assets that were used to build the dataset belongs to external sources which are listed in the github at `https://github.com/facebookresearch/PUG`. |
| **Does the dataset contain data that might be considered confidential?** | No. |
| **Does the dataset contain data that, if viewed directly, might be offensive, insulting, threatening, or might otherwise cause anxiety?** | No. |



**COLLECTION**



| | |
|---|---|
| **How was the data associated with each instance acquired?** | The data (3D assets) were acquired through the Unreal Engine Marketplace `https://www.unrealengine.com/marketplace/en-US/store` and Sketchfab `https://sketchfab.com/`. Assets were then incorporated into the Unreal Engine to generate realistic 3D scenes and corresponding images. The 3D assets were manually selected to ensure high quality. |
| **What mechanisms or procedures were used to collect the data?** | Manual human curation. Assets were manually collected. |
| **If the dataset is a sample from a larger set, what was the sampling strategy?** | For PUG: Animals and PUG: SPAR, all combinations are included. For PUG: ImageNet and PUG: AR4T, a random sample of possible combinations is provided. |
| **Who was involved in the data collection process and how were they compensated?** | Only the authors of this work were involved. |
| **Over what timeframe was the data collected?** | The data were collected between June 2022 and June 2023 |
| **Were any ethical review processes conducted?** | No. |
| **Did you collect the data from the individuals in question directly, or obtain it via third parties or other sources (e.g., websites)?** | Third parties: Unreal Engine Marketplace `https://www.unrealengine.com/marketplace/en-US/store` and Sketchfab `https://sketchfab.com/`. |
| **Were the individuals in question notified about the data collection? If so, please describe (or show with screenshots or other information) how notice was provided, and provide a link or other access point to, or otherwise reproduce, the exact language of the notification itself.** | There is no personally identifiable information in our datasets as they are purely synthetic and contain no images of people. We purchased 3D assets from different marketplaces where required, however we did not explicitly contact the individual creators. |
| **Did the individuals in question consent to the collection and use of their data? If so, please describe (or show with screenshots or other information) how consent was requested and provided, and provide a link or other access point to, or otherwise reproduce, the exact language to which the individuals consented.** | N/A. See above. |

| If consent was obtained, were the consenting individuals provided with a mechanism to revoke their consent in the future or for certain uses? If so, please provide a description, as well as a link or other access point to the mechanism (if appropriate). | N/A. See above. |
|---|---|
| Has an analysis of the potential impact of the dataset and its use on data subjects (e.g., a data protection impact analysis) been conducted? If so, please provide a description of this analysis, including the outcomes, as well as a link or other access point to any supporting documentation. | No data about specific individuals is included in these data. See above. |

### PREPROCESSING

| Was any preprocessing/cleaning/labeling of the data done? | N/A. |
|---|---|
| Was the "raw" data saved in addition to the preprocessed/cleaned/labeled data? | N/A. |
| Is the software that was used to preprocess/clean/label the data available? | N/A. |

### USES

| Has the dataset been used for any tasks already? | Yes, these data were used for the experiments that were presented in this paper. |
|---|---|
| Is there a repository that links to any or all papers or systems that use the dataset? | No. |
| What (other) tasks could the dataset be used for? | These Datasets could be used widely for evaluating and training neural networks. For example, assessing disentanglement of models with respect to PUG: Animals factors of variation (e.g. with DCI metric Eastwood and Williams [2018]). |
| Is there anything about the composition of the dataset or the way it was collected and preprocessed/cleaned/labeled that might impact future uses? | No. |
| Are there tasks for which the dataset should not be used? | These datasets **should not be used** for generative modelling purposes. |

### DISTRIBUTION

| Will the dataset be distributed to third parties outside of the entity on behalf of which the dataset was created? | Yes, the dataset will be publicly distributed. |
|---|---|
| How will the dataset will be distributed? | Tarball on a website. |
| Will the dataset be distributed under a copyright or other intellectual property (IP) license, and/or under applicable terms of use (ToU)? | The license of the dataset is **cc-by-nc with the mention that these data should not be used for generative AI purposes**. |
| Have any third parties imposed IP-based or other restrictions on the data associated with the instances? | See EpicGames. |

| | |
|---|---|
| **Do any export controls or other regulatory restrictions apply to the dataset or to individual instances?** | N/A |

| | |
|---|---|
| **Who will be supporting/hosting/maintaining the dataset?** | Meta AI. |
| **How can the owner/curator/manager of the dataset be contacted?** | Please contact the corresponding author of this paper. |
| **Is there an erratum?** | No. |
| **Will the dataset be updated?** | Yes the dataset will be updated with versioning. |
| **If the dataset relates to people, are there applicable limits on the retention of the data associated with the instances (e.g., were the individuals in question told that their data would be retained for a fixed period of time and then deleted)? If so, please describe these limits and explain how they will be enforced.** | N/A. |
| **Will older versions of the dataset continue to be supported/hosted/maintained? If so, please describe how. If not, please describe how its obsolescence will be communicated to dataset consumers.** | It depends. If the dataset is updated because one of the asset creators has requested to remove their assets, we will not continue to host the dataset containing these assets. Only the newer version of the dataset which will not contain these assets will be available. |
| **If others want to extend/augment/build on/contribute to the dataset, is there a mechanism for them to do so?** | No mechanisms are in place yet, but they can contact the authors of this paper if they would like to contribute. |

Table 4: **Datasheet for PUG**, following the framework introduced by Gebru et al. [2021].

## B.2 PUG: Animals

PUG Animals contains 215 040 pre-rendered images using 70 animals assets, 64 backgrounds, 3 sizes, 4 textures, under 4 different camera orientations. To create PUG: Animals, we use Animals assets from the following bundle in the Epic Game Marketplace ( `https://www.unrealengine.com/marketplace/en-US/product/complete-animals/reviews`). The list of environments used can be found in the dataset folder or at `https://github.com/facebookresearch/PUG`. Below, we list all the values for the factors of variation, we have used:

- **World_Name** : ["Egypt", "Desert", "AmusementPark", "ArcadeClub", "Arena", "Battleground", "Catacombs", "Tableland", "EuropeanStreet", "JunkYard", "OceanFloor", "Racetrack", "Ruins", "SciFiCity", "SciFiGarage", "SpaceIsland", "SpaceHangar", "SpatialStation", "TokyoDay", "TokyoNight", "TrainStation", "Bridge", "Beach", "BusStationInterior", "BusStationExterior", "Subway", "IndoorStairs", "Bar", "ScreeningCheckpoint", "Circus", "Appartment", "Hallway", "TrashRoom", "FuturisticSubway", "Footbridge", "BoxingRing", "Hangar", "Mansion", "ShoppingMall", "ConferenceRoom", "SpacePort", "VillageOutskirt","VillageSquare","Courtyard", "ElvenRuins", "Forge", "Library", "Museum", "Gallery", "ModernGallery", "Opera", "AncientAgora", "Restaurant", "RuralAustralia", "AustralianRoad", "ShadyRoad", "SaltFlats", "Castle", "StylizedEgypt", "Temple", "Snow", "Grass", "DryGrass", "Forest"],

- **Character_Name** : ["Goldfish","Caribou","Elephant","Camel","Penguin","Cassowary","Zebra", "Turtle","Bear","Beaver","Capybara","Crocodile","Armadillo","Cat","Gecko","Crow","GiantAnteater", "GiantTortoise","KomodoDragon","Rhinoceros","Dolphin","EarlessSeal","FruitBat","Goat","Hippopotamus", "Horse","Impala","Lion","Orca","Pig","Rabbit","Squirrel","Tapir","Wildbeest","Wolf","Anlylosaurus", "BlackRockFish","Parasaurolophus","PoisonDartFrog","Spinosaurus","Triceraptos","Chicken", "HarpyEagle","Ostrich","Raven","RedCrownedCrane","Robin","Seagull","Secretarybird","Shoebill","Swan",

"Toucan","Vulture","Ammonite","Ant","Scorpion","GoldBeetle","Hornet","SnowCrab","Tarantula","WhiteShark", "Tuna","Arowana", "Ayu", "Betta", "Koi", "Pirarucu", "Salmon", "Cattle", "Jerboa"],

- **Character_Scale** : [0.7, 1.0, 1.3],
- **Camera_Yaw** : [0, 45, 225, 315],
- **Character_Texture** : ["Default", "Sky", "Grass", "Asphalt"]

PUG: Animals is built by using all combinations of the factor of variation above. In Figure 7, we show random images from the PUG: animal dataset that highlight the diversity of this dataset.

Following Liu et al. [2023], we present a comparison in Table 5 with other datasets that are often used in OOD research:

| Image Data | Colored MNIST Arjovsky et al. [2020] | MNIST-R Ghifary et al. [2015] | Waterbirds Sagawa* et al. [2020] | Biased-Cars Madan et al. [2022] | Nico++ Xingxuan Zhang [2022] | PUG: Animals Ours |
|---|---|---|---|---|---|---|
| # Domains | 3 | 6 | 2 | - | 10 | 64 |
| # Categories | 2 | 10 | 2 | 5 | 80 | 70 |
| # Examples | - | 6k | 4.8k | 450k | 230k | 215k |
| Shift Type | Color | Angle | Background | Views | Background | Back./Text./Size./View |
| Image Type | Digits | Digits | Birds | Synthetic Cars | Real Objects | Synthetic Animals |

Table 5: Comparing PUG: Animals with other datasets traditionally used for OOD research. In contrast to other datasets that have variations across only a single domain, that have noisy annotations or that are to unrealistic, PUG: Animal over high quality images with reliable annotations across different domains such as the background,texture,size and view.

## B.3 PUG: ImageNet

PUG: ImageNet contains 88,328 pre-rendered images using 724 assets representing 151 ImageNet classes with 64 backgrounds, 7 sizes, 10 textures, 18 different camera orientation, 18 different character orientation and 7 light intensity. Below is the values we used for each of these factors:

- **World_Name** : ["Egypt", "Desert", "AmusementPark", "ArcadeClub", "Arena", "Battleground", "Catacombs", "Tableland", "EuropeanStreet", "JunkYard", "OceanFloor", "Racetrack", "Ruins", "SciFiCity", "SciFiGarage", "SpaceIsland", "SpaceHangar", "SpatialStation", "TokyoDay", "TokyoNight", "TrainStation", "Bridge", "Beach", "BusStationInterior", "BusStationExterior", "Subway", "IndoorStairs", "Bar", "ScreeningCheckpoint", "Circus", "Appartment", "Hallway", "TrashRoom", "FuturisticSubway", "Footbridge", "BoxingRing", "Hangar", "Mansion", "ShoppingMall", "ConferenceRoom", "SpacePort", "VillageOutskirt","VillageSquare","Courtyard", "ElvenRuins", "Forge", "Library", "Museum", "Gallery", "ModernGallery", "Opera", "AncientAgora", "Restaurant", "RuralAustralia", "AustralianRoad", "ShadyRoad", "SaltFlats", "Castle", "StylizedEgypt", "Temple", "Snow", "Grass", "DryGrass", "Forest"],
- **Character_Name** : 724 Sketchfab assets (See github for the list)
- **Character_Rotation_Yaw** : [0, 45, 135, 180, 225, 270],
- **Character_Rotation_Roll** : [45, 90, 135, 180, 225, 270],
- **Character_Rotation_Pitch** : [45, 90, 135, 180, 225, 270],
- **Character_Scale** : [0.5, 0.6, 0.7, 0.8, 1.3, 1.6],
- **Camera_Roll** : [45, 90, 135, 180, 225, 270],
- **Camera_Pitch** : [240, 260, 280, 300, 320, 340],
- **Camera_Yaw** : [0, 45, 135, 180, 225, 270],
- **Character_Texture** : ["Default", "Sky", "Green", "Gray", "Red", "Grass", "Color", "Black", "Curtain"],,
- **Scene_Light** : ["255,255,255,0","0,0,255,0", "0,255,0,0", "255,0,0,0", "0,255,255,0", "255,0,255,0", "255,255,0,0"] (The value for the lights are in RGBA format)

To generate PUG:ImageNet, we change only one factor at a time for each assets. When changing the background (World_Name), all the other factors (Camera/Object Orientation, Size, Texture, Light)

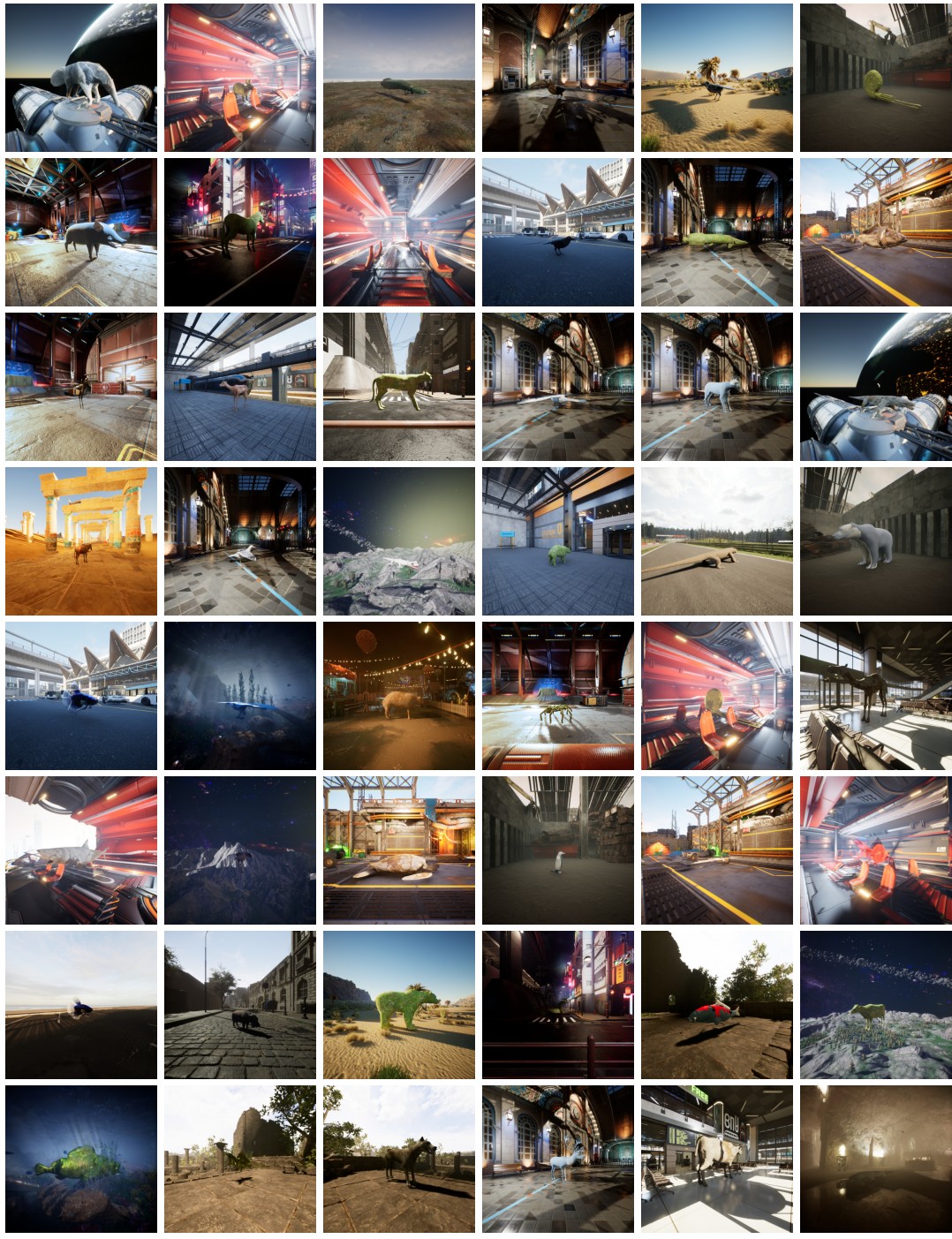

Figure 7: Random images taken from the PUG: Animals dataset.

are at 0 or at their default value. When changing the other factors (Camera/Object Orientation, Size, Texture), the background is set to "SaltFlats_0" (Which is the most *basic* background). When changing the light of the scene, we have used the environment "Opera".

The assets in PUG:ImageNet were selected base on 151 ImageNet classes which are listed below: ['BirdHouse', 'Chest', 'Bagel', 'WarPlane', 'Rocking_Chair', 'Bridge', 'Street_Sign', 'Cabbage', 'Pay_Phone', 'Butternut_Squash', 'JellyFish', 'Jack_O_Lantern', 'Bookcase', 'Stonewall', 'Punching_Bag', 'Toaster', 'Mushroom', 'Frog', 'Jeep', 'Television', 'Pineapple', 'Vacuum', 'Torch', 'Carousel', 'Desk', 'WineBottle', 'Wallet',

'Dining_Table', 'Military_uniform', 'Car_Wheel', 'Table_Lamb', 'Digital_Watch', 'Electric_Fan', 'Sweat-shirt', 'Komodo_dragon', 'Racket', 'Cheeseburger', 'Can_Opener', 'Pomegranate', 'Convertible', 'Laptop', 'Chicken_hen', 'Wolf', 'Bulletproof_vest', 'Shield', 'Bathtub', 'Throne', 'Lighter', 'Bycicle', 'Cofee_Mug', 'Motor_Scooter', 'Jean', 'Soccer_Ball', 'Vending_machine', 'Hatchet', 'Umbrella', 'Bear', 'Artichoke', 'Vase', 'Radiator', 'SpaceShuttle', 'Manhole_Cover', 'Polaroid_Camera', 'Traffic_Light', 'Radio', 'Soup_Bowl', 'Zuc-chini', 'Barrel', 'Tennis_Ball', 'Sunglasses', 'Microwave', 'Joystick', 'Aircraft_Carrier', 'Fox', 'Submarine', 'BasketBall', 'Running_Shoe', 'Chain-saw', 'Piano', 'Crate', 'Loupe', 'Minivan', 'Shirt', 'Remote_controler', 'Airliner', 'Sock', 'Shovel', 'Mask', 'Tractor', 'Sandal', 'Wooden_Spoon', 'Drum', 'Goldfish', 'Gasmask', 'Mailbox', 'Volley_Ball', 'Banana', 'Penguin', 'Sliding_Door', 'Pool_Table', 'Burrito', 'Candle', 'Purse', 'Canoe', 'Typewriter_Keyboard', 'Espresso_maker', 'Carton', 'Park_Bench', 'Screen', 'African_crocodile', 'Cat', 'Hay', 'Elephant', 'WaterBottle', 'Modem', 'Palace', 'Ice_Cream', 'Washer', 'Sewing_Machine', 'HairDryer', 'Rabbit', 'Dishwasher', 'Bell_Pepper', 'Ambulance', 'French_Loaf', 'Refrigerator', 'Mouse', 'Obelisk', 'Starfish', 'Brocolli', 'Microphone', 'Great_white_shark', 'Power-drill', 'Locomotive', 'Perfume', 'Whale', 'Screwdriver', 'Dial_telephone', 'Backpack', 'Harmonica', 'Binocular', 'Skirt', 'Pizza', 'Cowboy_Hat', 'Computer_Keyboard', 'Kangarou', 'Baseball', 'Tile_Roof', 'Lawn_Mower', 'Safe', 'Cellular_telephone']

In Figure 8, we show random images taken from the PUG: ImageNet dataset.

## B.4 PUG: SPAR

PUG: SPAR contains 43,560 pre-rendered images using 32 animals assets, 10 backgrounds, 4 positions and 4 textures. In contrast with the other PUG datsets, PUG: SPAR contain up to two animal in a single scene. For generating the PUG: SPAR dataset, we utilize the same subset of assets as PUG: Animals.

- **World_Name** : [ 'Desert', 'Arena', 'OceanFloor', 'Racetrack', 'TokyoDay', 'Circus', 'BoxingRing', 'AustralianRoad', 'SaltFlats', 'Museum'],

- **Character_Name** : ['Goldfish', 'Caribou', 'Elephant', 'Camel', 'Penguin', 'Zebra', 'Bear', 'Beaver', 'Cattle', 'Armadillo', 'Gecko', 'Crow', 'Scorpion', 'GiantTortoise', 'Tarantula', 'Rhinoceros', 'Dolphin', 'EarlessSeal', 'Goat', 'Hippopotamus', 'Horse', 'Impala', 'Lion', 'Orca', 'Pig', 'Rabbit', 'Squirrel', 'Chicken', 'WhiteShark', 'Anlylosaurus', 'BlackRockFish', 'PoisonDart-Frog'],

- **Character_pos** : ["Left/Right", "Bottom/Top"]

- **Character_Texture** : ["Default", "Blue/Red", "Grass/Stone"]

In Figure 8, we show random images taken from the PUG: SPAR.

## B.5 PUG: AR4T

**Assets and Environments** For generating the PUG: AR4T dataset, we utilize the same subset of Sketchfab assets as used in PUG: ImageNet. This leaves us with a set of 680 unique assets chosen from across 151 ImageNet categories, each manually inspected to quality control for photo-realism as much as possible. For this PUG dataset we are primarily concerned with object-object and object-attribute information, hence we utilize single camera and character orientation. Since Sketchfab assets differs widely in their scales, we first scaled down the longest edge of the asset bounding box to 150 pixels to normalize the order of magnitude of the asset dimensions, before any further scaling based on attributes. Next, we select a total of 26 unique environments as our background environments. The richness of some environments enables us to manually select different camera views in each environment as a novel environment view, and we generate a total of 28 visually unique backgrounds for our PUG: AR4T dataset. We provide each background with human-intelligible descriptive names for use in the dataset captions (Table 6). The PUG: AR4T dataset is composed with two subset described in the following sections.

**PUG: AR4T-Relations** For generating the PUG: AR4T-Relations subset, we utilize the spatial relationships from Visual Genome that are not symmetric, as noted in the ARO benchmark Yuksekgonul et al. [2023]. The set of relationships used in PUG: AR4T-Relations is given in Table 7. It consists of three unary relations (at, in, inside) and ten binary relations (above, on, on top of, behind, in front of, below, beneath, under, to

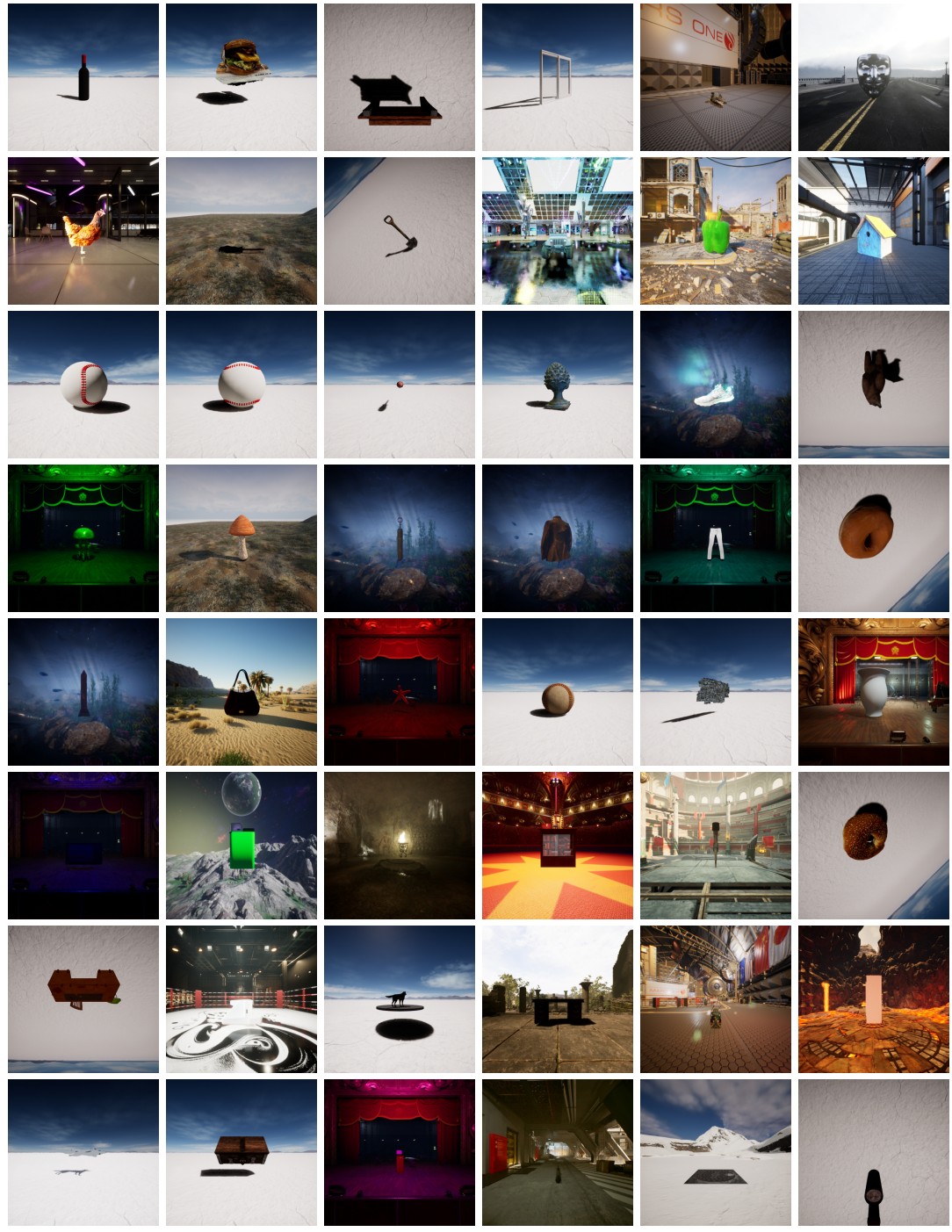

Figure 8: Random images taken from the PUG: ImageNet dataset.

the left of, to the right of.) For each relation, the corresponding objects are picked randomly from the set of assets, and placed in the scene based on the co-ordinates given in Table 7. We add a random offset in the range [0-25] along each dimension for every individual object for every scene, so that the dataset does not contain shortcuts where the object locations can directly inform the underlying relation. The size of each asset is chosen randomly on a scale of 1-10, where 1 corresponds to 110 pixels and 10 to 200 pixels for the longest edge of the scaled asset.

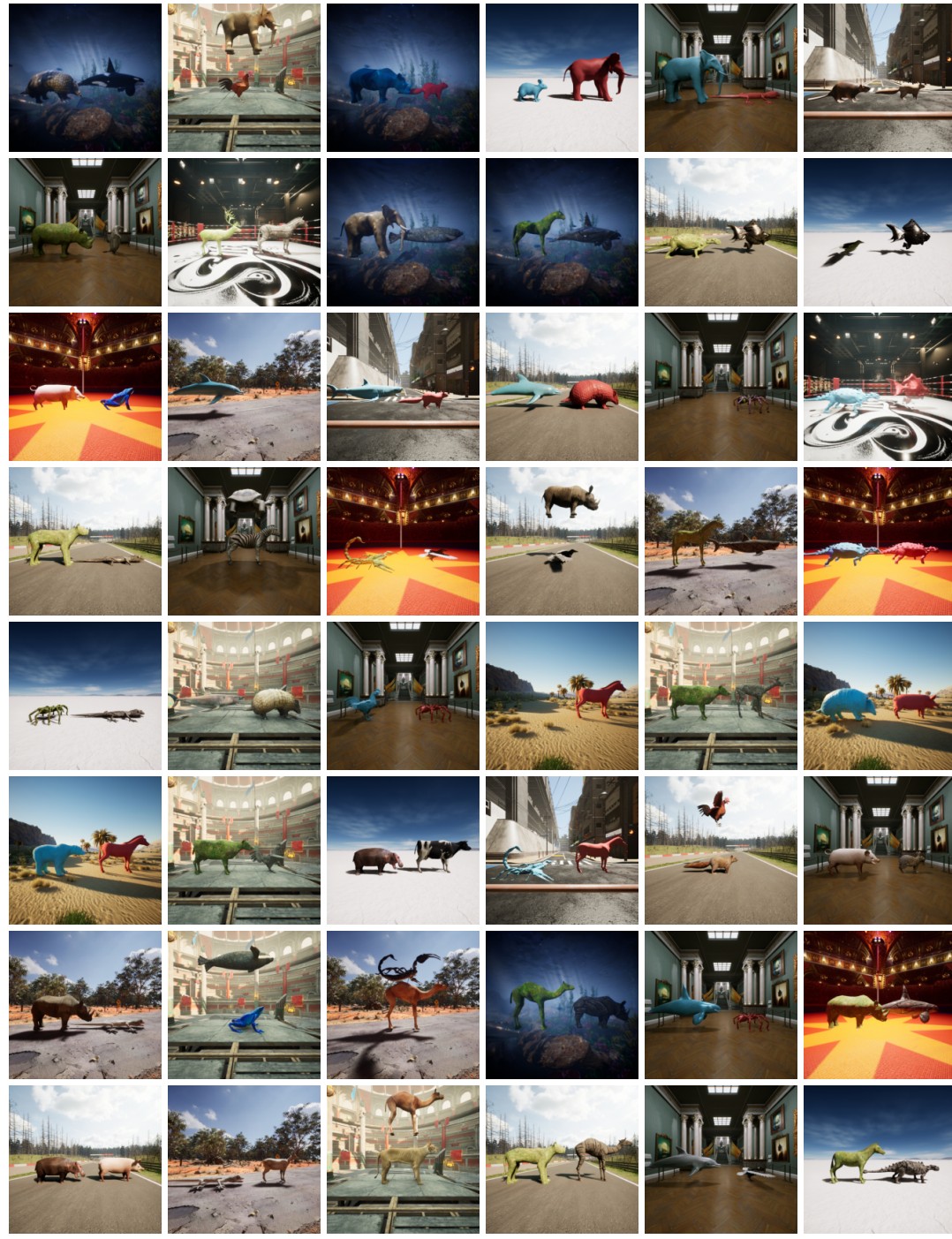

Figure 9: Random images taken from the PUG: SPAR dataset.

The caption for a scene is generated using one of two templates based on whether the spatial relationship in the scene is unary or binary:

- **Unary Relation:** *"[ImageNet class label of asset 1] [relation] [human-intelligible background description]"* e.g. Banana inside museum

| Environment Name (with camera view variant) | Descriptive Caption for PUG | Environment Name (with camera view variant) | Descriptive Caption for PUG |
|---|---|---|---|
| Arena | arena | IceRoad | icy road |
| VillageOutskirt | village | Jungle | jungle |
| VillageSquare | village | Library | library |
| Battleground | battleground | Museum | museum |
| Beach | beach | OceanFloor | ocean floor |
| Bridge | bridge | AncientAgora | castle outskirts |
| Circus | circus | Racetrack | race track |
| Cliff | cliff | RuralAustralia | rural wilderness |
| Courtyard | courtyard | AustralianRoad | desert road |
| Egypt | egypt | SaltFlats | salt flats |
| ElvenRuins | ruins | SpaceIsland | space |
| EuropeanStreet | european street | StylizedEgypt | egypt |
| FightingArena | fighting arena | Temple | temple |
| Forge | forge | TrainStation | train station |

Table 6: Environments and descriptive captions used in PUG: Attributes & Relations

| Relation | Object 1 coordinates (wrt origin) | Object 2 coordinates (wrt origin) | Negative Relation |
|---|---|---|---|
| At | (0, 0, 0) | N/A | N/A |
| In | (0, 0, 0) | N/A | N/A |
| Inside | (0, 0, 0) | N/A | N/A |
| Above | (0, 0, 300) | (0, 0, 0) | Below, Beneath, Under |
| On top of | (0, 0, 300) | (0, 0, 0) | Below, Beneath, Under |
| On | (0, 0, 300) | (0, 0, 0) | Below, Beneath, Under |
| Below | (0, 0, 0) | (0, 0, 300) | Above, On Top Of, On |
| Beneath | (0, 0, 0) | (0, 0, 300) | Above, On Top Of, On |
| Under | (0, 0, 0) | (0, 0, 300) | Above, On Top Of, On |
| Behind | (150, 0, 0) | (-150, 0, 0) | In front of |
| In front of | (-150, 0, 0) | (150, 0, 0) | Behind |
| To the left of | (0, -150, 0) | (0, 150, 0) | To the right of |
| To the right of | (0, 150, 0) | (0, -150, 0) | To the left of |

Table 7: Relations, corresponding asset locations wrt to camera origin, and corresponding hard negative relation used in PUG: AR4T

- **Binary Relation:** *"[ImageNet class label of asset 1] [relation] [ImageNet class label of asset 2] [(random) unary relation] [human-intelligible background description]"* e.g. Banana to the left of chair inside museum

For each binary relation, we also sample the corresponding hard negative scene and caption, by replacing the relation with its negative from Table 7 such that the semantic meaning of the scene and caption represents a switch from the original relation between objects. For example, the negative of *"Banana to the left of chair inside museum"* is given by *"Banana to the right of chair inside museum"*.

We sample a total of 310 binary relation scenes (155 random + 155 negatives), and 310 unary relation scenes for each background, thus leading to a dataset of 112,840 image-caption samples (28 backgrounds x (310 pairs x 10 binary relations + 310 pairs x 3 unary relations)). The PUG:Relations dataset generation process described above is summarized as psueuocode in Algorithm 1.

Lastly, we split the dataset into 101,920 train and 10,920 test image-caption samples, such that the test set is balanced by background and relations (28 backgrounds x 13 relations x 30 samples). We also release the subset of training and test samples that only contain pairs of objects in scenes along with their hard negatives, which contains 78,400 training and 8,400 test samples, or 39,200 training pairs and 4,200 test pairs. This subset enables Winoground Thrush et al. [2022] style evaluation as well as training with hard visual negative mining for VLMs in future work. The PUG:Relations dataset generation process described above is summarized as psueuocode in Algorithm 1

**PUG: AR4T-Attributes** For PUG: AR4T-Attributes the selection of assets, relations between assets, and spatial locations of assets is done exactly as for PUG: AR4T-Relations. However, since the focus of this dataset is on the object-attribute pairs, the relations between objects are not represented in the corresponding scene caption in any form. The attribute for each object is chosen randomly from the set of 53 attributes described in Table 8. For size based attributes, the asset's material instance remains the same but its size is varied between 50 pixels (`short`, `small`, `little`, `tiny`) and

**Algorithm 1** PUG: AR4T-Relations subset generation

Unary = {At, In, Inside}
Categories = {Set of ImageNet Class Labels}
Environments = {Set of Unreal Environments}
Relations = {Set of Relations}
NegRelations = {Dictionary of relations as key, semantically negative relations as values}
Assets = {Dictionary of Sketchfab assets, keys being ImageNet Class Labels}
AssetLocations = {Function that returns locations of assets based on relation with a random offset}
Dataset = $\phi$
**for** env $in$ Env **do**
    **for** rel $in$ Rel **do**
        ▷ For binary relations we also add the negative scene+caption to the dataset, thus
        the effective number of samples per background is 2*15 = 30
        **if** rel $\in$ Unary **then**
            num_samples = 15
        **else**
            num_samples = 30
        **end if**
        **for** $i = 1$ to num_samples **do**
            cat1 = random.choice(Categories)
            asset1 = random.choice(Assets[cat1])
            **if** rel $\notin$ Unary **then**
                cat2 = random.choice(Categories)
                asset2 = random.choice(Assets[cat2])
            **end if**
            location1, location2 = AssetLocations(rel)
            scene1 = TorchMultiverse(asset1, asset2, location1, location2, env)
            **if** rel $\notin$ Unary **then**
                rel2 = random.choice(Unary)
                caption1 = cat1 + ' ' + rel + ' ' + cat2 + ' ' + rel2 + ' ' + env
                Dataset $\cup$ {scene1, caption1}
                ▷ Generate hard negative scene and caption
                scene1 = TorchMultiverse(asset1, asset2, location2, location1, env)
                caption2 = cat1 + ' ' + NegRelations[rel] + ' ' + cat2 + ' ' + rel2 + ' ' + env
                Dataset $\cup$ {scene2, caption2}
            **else**
                caption1 = cat1 + ' ' + rel + ' ' + env
                Dataset $\cup$ {scene1, caption1}
            **end if**
        **end for**
    **end for**
**end for**
**return** Dataset

---

200 pixels (`big, long, tall, large`). For all other attributes, the material instance of the object is changed in Unreal using Pytorch Multiverse.

The caption of the scene is generated using one of two templates based on whether the spatial relationship in the scene is unary or binary:

- **Unary Relation:** *"[Attribute of asset 1] [ImageNet class label of asset 1] [relation] [human-intelligible background description]"* e.g. Green banana inside museum

- **Binary Relation:** *"[Attribute of asset 1][ImageNet class label of asset 1] and [Attribute of asset 2][ImageNet class label of asset 2] [(random) unary relation] [human-intelligible background description]"* e.g. Green banana and large chair inside museum

For each binary relation, we also sample the corresponding hard negative scene and caption, by swapping the attributes between objects such that the semantic meaning of the scene and caption

| Attribute Category | Attribute | Variants | Attribute Category | Attribute | Variants |
|---|---|---|---|---|---|
| Material | Brick | 3 | Color | Black | 2 |
| | Metal | 2 | | Blue | 2 |
| | Wood | 2 | | Yellow | 2 |
| | Glass | 2 | | White | 2 |
| | Cloth | 2 | | Green | 2 |
| | Plastic | 2 | | Gray | 2 |
| | Rock | 2 | | Brown | 2 |
| Size | Big | 1 | | Red | 2 |
| | Long | 1 | | Silver | 2 |
| | Tall | 1 | | Orange | 2 |
| | Large | 1 | | Pink | 2 |
| | Short | 1 | | Gold | 2 |
| | Small | 1 | Texture | Striped | 2 |
| | Little | 1 | | Dark | 2 |
| | Tiny | 1 | | Cloudy | 2 |
| | | | | **Total=** | 53 |

Table 8: Attributes and corresponding number of variants used in PUG: AR4T-Attributes

represents a switch from the original object-attribute associations. For example, the negative of *"Green banana and large chair inside museum"* is given by *"Large banana and green chair inside museum"*. For each of the 53 object attributes, we sample 2 scenes of it in conjugation with another object and attribute, and 2 scenes of the attribute in isolation in a scene. We repeat this for each possible background, thus leading to a dataset of 160,272 image-caption samples (28 backgrounds x (53 attributes x (53 attributes x 2 samples) + 2 samples)). The PUG:Attributes dataset generation process described above is summarized as psueduocode in Algorithm 2.

Lastly, we split the dataset into 147,976 train and 12,296 test images. For each attribute pair in a scene, we select 4 image-caption samples in the test set (4 samples x 53 attributes x 53 attributes = 11,236 sample). And we sample the remaining test samples by selecting 20 image-caption samples for each attribute in isolation in a scene (20 samples x 53 attributes = 1,060 samples). Similar to PUG-attributes, we also separately release the subset of training and test set with scenes and captions containing only pairs of scenes with their corresponding hard negatives, leading to 146,158 training samples and 11,236 test samples, or 73,0739 training and 5,618 test sample pairs.

**Caption variants in PUG: AR4T**    In order to emphasize the idea that each scene can have multiple descriptive captions associated with it, we utilize a simple template to generate alternate captions for scenes with binary relations that are semantically consistent but linguistically different. During fine-tuning, the model randomly sees either the original caption or the alternate caption. The presence of alternate captions also prevents the VLM to learn shortcuts between the position of the object descriptions in captions and the underlying spatial relationship or object-attribute association.

- **PUG: Relations** *"[ImageNet class label of asset 2] [negative relation] [ImageNet class label of asset 1] [(random) unary relation] [human-intelligible background description]"* e.g. The alternate caption for 'Banana to the left of chair inside museum' is 'Chair to the right of banana inside museum'

- **PUG: Attributes** *"[Attribute of asset 2][ImageNet class label of asset 2] and [Attribute of asset 1][ImageNet class label of asset 1] [(random) unary relation] [human-intelligible background description]"* e.g. The alternate caption for 'Green banana and large chair inside museum' is 'Large chair and green banana inside museum'

**Algorithm 2** PUG: AR4T-Attributes subset generation

---

Unary = {At, In, Inside}
Categories = {Set of ImageNet Class Labels}
Environments = {Set of Unreal Environments}
Relations = {Set of Relations}
Attributes = {Set of Attributes}
Assets = {Dictionary of Sketchfab assets, keys being ImageNet Class Labels}
AssetLocations = {Function that returns locations of assets based on relation with a random offset}
Dataset = $\phi$
**for** att1 $in$ Attributes **do**
    **for** att2 $in$ Attributes + [None] **do**
        **if** att2 == None **then**
            num_samples = 20
        **else**
            ▷ For binary relations we also add the negative scene+caption to the dataset, thus the effective number of samples per background and attribute is 2*2 = 4
            num_samples = 2
        **end if**
        **for** $i = 1$ to num_samples **do** env = random.choice(Environments)
            **if** att2 == None **then**
                rel = random.choice(Unary)
            **else**
                rel = random.choice(Relations - Unary)
            **end if**
            cat1 = random.choice(Categories)
            asset1 = random.choice(Assets[cat1])
            **if** att2 != None **then**
                cat2 = random.choice(Categories)
                asset2 = random.choice(Assets[cat2])
            **end if**
            location1, location2 = AssetLocations(rel)
            scene1 = TorchMultiverse(asset1, asset2, location1, location2, att1, att2, env)
            **if** rel $\notin$ Unary **then**
                rel2 = random.choice(Unary)
                caption1 = att1 + ' ' + cat1 + ' and ' + att2 + ' ' + cat2 + ' ' + rel2 + ' ' + env
                Dataset ∪ {scene1, caption1}
                ▷ Generate hard negative scene and caption by switching object-attribute associations
                scene2 = TorchMultiverse(asset1, asset2, location1, location2, att2, att1, env)
                caption2 = att2 + ' ' + cat1 + ' and ' + att1 + ' ' + cat2 + ' ' + rel2 + ' ' + env
                Dataset ∪ {scene2, caption2}
            **else**
                caption1 = att1 + ' ' + cat1 + ' ' + rel + ' ' + env
                Dataset ∪ {scene1, caption1}
            **end if**
        **end for**
    **end for**
**end for**
**return** Dataset

---

# C  Additional experimental details

## C.1  Equivariance study details

In section 3.2, we used PUG: Animals to study how foundation vision-language models behave with respect to changes in factors of variations. We showed high image and text equivariance with respect to background, and text equivariance with respect to size and texture too. Here, we provide more details and results.

In our study, we use the following pretrained models:

- BLIP with ViT base backbone from the Huggingface transformers library [Wolf et al., 2020], trained on COCO dataset [Lin et al., 2014],
- NegCLIP from [Yuksekgonul et al., 2023],
- As in [Yuksekgonul et al., 2023] we use X-VLM pretrained on COCO dataset from `https://github.com/zengyan-97/X-VLM`,
- Flava with ViT-B/32 backbone from Huggingface transformers (`https://huggingface.co/facebook/flava-full`)
- CLIP models all come from OpenAI CLIP `https://github.com/openai/CLIP`. We use the versions with ResNet50, ResNet101, ViT-L/14, ViT-B/16, ViT-B/32.

We compute equivariance to each of the factors of variations. Since the text captions do not take into account camera and asset orientations, when we compute the equivariance with respect to the other factors we take only samples for a given orientation of the character and camera (0 for roll, pitch and yaw in both cases). Furthermore, in the text caption we replace sizes by three adjectives as follow: 0.7 is mapped to small, 1.0 to medium and 1.3 to big. Inspired by [Bouchacourt et al., 2021], we compute equivariance as the alignment between embedding difference vectors. That is, we compute embedding difference vectors $z_i - z_j$ where $z_i$ and $z_j$ are the (normalized) embeddings of two images (or texts) of an object undergoing a given factor change. We then measure pairwise alignment as cosine similarity of embedding difference vectors (either between images pairs, text pairs or image-text pairs) corresponding to the same factor change. We report averaged cosine similarity of randomly paired vectors, and a higher value implies higher equivariance.

Note that a model can present image equivariance but no text equivariance (caption and image are not guaranteed to be encoded in the same vector), or have high equivariance across modalities but no image or text equivariance and vice-versa. In Figure 11 we report image equivariance with

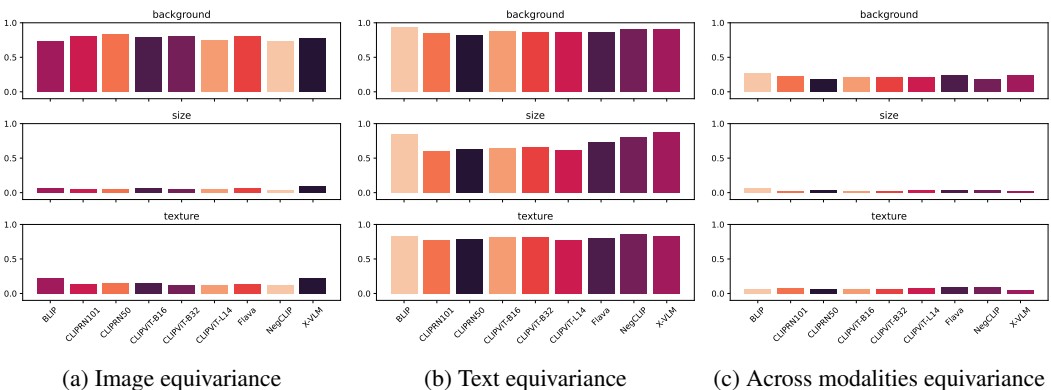

(a) Image equivariance      (b) Text equivariance      (c) Across modalities equivariance

Figure 10: Measuring foundation models equivariance thanks to PUG: Animals: all three factors.

respect to orientation of the camera yaw. We see that there is little to no equivariance to it, suggesting that image embeddings are more predictable when changing background than the camera yaw. Note that foundation models representations belong to the hypersphere, yet our measurement of equivariance as parallelism (measured with cosine similarity) relies on Euclidean geometry. Still, cosine similarity is a starting point to showcase how PUG: Animals can be used to study models'

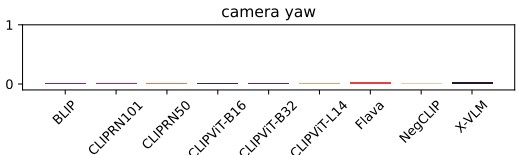

Figure 11: Additional image equivariance results with respect to camera yaw.

representational spaces. This could also explain the higher equivariance values of text representations: since textual captions follow the same template, embeddings might be close to each other (relative to image embeddings distances). In this case the hypersphere locally behaves like an Euclidean space [Lee, 2000], for which the Euclidean geometry is better suited. We leave for future work the exploration more complex equivariance metrics potentially based on spherical geometry to study foundation models' representational spaces. Studying model's representations is key to better understanding and improving them Bouchacourt et al. [2021], Xie et al. [2022], Ushio et al. [2021], Lenc and Vedaldi [2018]. Our study showcases that PUG: Animals advantages (rich diversity of factors, knowledge of their values, control either one factor at a time but also all together) make it a great dataset to study state-of-the-art models representational properties.

## C.2 Classification with held out sets

In section 3.2, we study how one can leverage PUG: Animsl to study OOD generalization in two settings: 1) Generalization on unseen factors 2) Generalization on unseen combination of factors.

For this experiment, we use PUG: Animals with held out sets. Typically, we random select a number of number of animals (0, 10 or 20) within our 70 assets. Then for the remaining animals we decided to remove from PUG: Animals a number of background, object size or object texture. This give us a training set. Then the images that were excluded from this training set are put as a test or held out set in which we measure the performance of a model. This model is typically a Resnet50 trained for 100 epochs with AdamW as optimizer with a batch size of 2048.

## C.3 Robustness of SOTA models additional details

In addition to evaluating robustness for the models in the main paper, in Table 9 we provide an analysis of several additional models including the recent self-supervised DINOv2 model as well as BLIP a contrastive vision-language model. Parenthesis indicate the pretraining dataset: ImageNet 21k, LVD-142M, JFT 300M, LAION 400M, and LAION 2B. Models without parethesis are pretrained on the standard ImageNet-1k dataset. For ResNet models we use the publicly available pretrained checkpoints from the Timm package based on the training recipe from Wightman et al. [2021]. For the vision transformer models and Swin we use the pretrained models from the Timm package with patch size 16 for ViT and Swin Base with a patch size of 4 and window size of 7. For the BiT model we use the pretrained checkpoint trained on Google's JFT 300M dataset from the Timm package with a ResNetv2 101 architecture. For DINOv2, we use the officially released repo to evaluate the base ViT architecture trained on 132 million samples [Oquab et al., 2023]. For BLIP we use the checkpoint available in HuggingFace and evaluate the model using zero shot classification via the prompt 'This is a photo of a [ ]'. We evaluate CLIP model variants in a similar zero shot fashion and rely on OpenCLIP's implementation. The parenthesis indicate the pretraining dataset size from the LAION dataset. We report the average accuracy as each factor (see columns of Table 9) varies.

We also measure the relationship between standard in-distribution accuracy and robustness based on the accuracy as each factor in PUG:ImageNet varies. We measure Pearson's correlation coefficient between ImageNet accuracy and accuracy for each factor in Table 10. We find no statistically significant relationship between standard classification accuracy and factor robustness.

### C.3.1 Performances

To understand if the differences in performance between the real ImageNet and our PUG dataset is caused by a sim-to-real gap or by the factor of variations, we show below the zero-shot accuracy obtained with a pretrained resnet101 for each class in PUG: ImageNet. There is only 3 classes for

| | PUG: ImageNet | | | | | | | | | | |
|---|---|---|---|---|---|---|---|---|---|---|---|
| | ImageNet | Camera_Yaw | Camera_Pitch | Camera_Roll | Object_Yaw | Object_Pitch | Object_Roll | Object_Scale | Object_Texture | Scene_Light | Background |
| ResNet50 | 81.5 | 38.1 | 33.1 | 26.9 | 38.0 | 23.6 | 22.9 | 35.7 | 27.0 | 13.6 | 29.5 |
| ResNet101 | 82.3 | 43.4 | 35.9 | 29.4 | 45.1 | 26.7 | 25.6 | 39.7 | 31.1 | 14.1 | 32.8 |
| BiT (JFT300M) | 80.3 | 40.5 | 32.3 | 26.0 | 42.1 | 23.6 | 22.8 | 37.3 | 23.4 | 6.3 | 20.5 |
| DINOv2 (LVD-142M) | 84.5 | 45.6 | 41.1 | 37.4 | 47.5 | 28.8 | 28.5 | 43.1 | 35.0 | 6.1 | 30.9 |
| Flava (PMD 70M) | 75.5 | 31.7 | 23.4 | 17.6 | 30.8 | 17.6 | 15.4 | 30.5 | 24.2 | 7.8 | 21.9 |
| Swin | 83.6 | 56.0 | 45.6 | 41.8 | 56.9 | 35.3 | 34.2 | 52.9 | 40.1 | 19.1 | 42.0 |
| ViT-Base | 84.3 | 37.5 | 34.3 | 31.7 | 38.0 | 21.8 | 20.5 | 33.0 | 28.5 | 4.1 | 26.6 |
| ViT-Large | 85.8 | 52.2 | 40.4 | 37.1 | 52.4 | 30.4 | 28.4 | 46.4 | 42.9 | 8.9 | 34.6 |
| BLIP (100+M) | – | 0.5 | 0.4 | 0.5 | 0.8 | 0.6 | 0.7 | 0.9 | 0.7 | 0.7 | 0.7 |
| CLIPViTB32 (2B) | 66.6 | 44.0 | 31.5 | 24.1 | 43.8 | 24.8 | 21.8 | 42.2 | 34.7 | 3.3 | 26.0 |
| CLIPViTB32 (400M) | 62.9 | 41.7 | 30.2 | 22.1 | 41.6 | 23.8 | 20.9 | 40.1 | 34.4 | 5.7 | 24.4 |
| CLIPViTL14 (2B) | 75.3 | 49.7 | 34.9 | 28.2 | 50.3 | 26.3 | 25.3 | 46.8 | 39.4 | 4.8 | 30.8 |
| CLIPViTL14 (400M) | 72.8 | 52.3 | 39.8 | 35.7 | 51.8 | 29.0 | 26.4 | 50.6 | 41.1 | 4.3 | 33.0 |

Table 9: Robustness measured by average accuracy across factors. We report zero shot classification accuracy for BLIP, Flava, all CLIP models. The pretraining dataset is indicated in parenthesis next to each model name with ImageNet-1k being the default unless otherwise indicated.

| factor | correlation | pvalue |
|---|---|---|
| Object Pitch | 0.29 | 0.45 |
| Camera Roll | 0.61 | 0.08 |
| Camera Pitch | 0.53 | 0.14 |
| Camera Yaw | 0.12 | 0.76 |
| Background | 0.43 | 0.25 |
| Object Yaw | 0.18 | 0.64 |
| Object Texture | -0.10 | 0.81 |
| Scene Light | 0.53 | 0.14 |
| Object Scale | -0.05 | 0.90 |
| Object Roll | 0.45 | 0.22 |

Table 10: We compute the correlation between standard ImageNet classification and robustness based on the accuracy for each factor. We find no statistically significant relationship for standard classification and factor robustness.

which there is not a single configurations of the factors that lead to a correct classification. For all the other classes, there is always at least one configuration for which the network is correctly predicting the class. In that instance, we assume that if the objects in a given class are correctly predicted at least one time, the failures in predicting the correct class for the same objects with different factors is probably due to the changes of factors.

**Zero-shot top-1 accuracy per class** Soccer_Ball: 100.00 , Pineapple: 95.62 , Barrel: 95.00 , Cellular_telephone: 89.45 , Pomegranate: 88.12 , Jack_O_Lantern: 85.94 , Vase: 85.62 , BirdHouse: 81.88 , BasketBall: 81.25 , Sewing_Machine: 80.94 , Umbrella: 80.00 , Washer: 78.75 , Pool_Table: 76.88 , Baseball: 76.88 , Safe: 76.25 , Cabbage: 75.78 , Cofee_Mug: 75.31 , Mask: 74.06 , Brocolli: 73.44 , Starfish: 72.50 , Rocking_Chair: 71.25 , Punching_Bag: 70.94 , Chicken_hen: 69.38 , WineBottle: 66.88 , Gasmask: 66.56 , Joystick: 64.06 , Television: 63.44 , Chest: 63.44 , Elephant: 62.50 , Bell_Pepper: 61.46 , Cheeseburger: 60.62 , Pay_Phone: 60.00 , Tennis_Ball: 58.44 , Jean: 56.25 , Binocular: 55.86 , Racket: 55.62 , Motor_Scooter: 55.00 , Hay: 54.06 , Park_Bench: 53.44 , Bookcase: 53.44 , Zucchini: 52.08 , Banana: 50.00 , Sliding_Door: 48.75 , Military_uniform: 47.50 , Ambulance: 47.50 , Pizza: 47.19 , Tractor: 46.88 , Dishwasher: 46.88 , Cowboy_Hat: 46.35 , Drum: 45.31 , Typewriter_Keyboard: 44.92 , Toaster: 44.69 , Obelisk: 44.38 , Laptop: 44.38 , Throne: 43.75 , Backpack: 43.44 , Shield: 41.88 , Artichoke: 41.80 , Penguin: 41.56 , Bathtub: 40.31 , WaterBottle: 40.00 , SpaceShuttle: 37.19 , Bagel: 36.88 , Bear: 36.25 , Vacuum: 35.94 , Radiator: 35.62 , Shovel: 35.55 , Refrigerator: 35.00 , Running_Shoe: 34.38 , Goldfish: 34.38 , Crate: 34.38 , Polaroid_Camera: 33.98 , Table_Lamp: 33.75 , Bulletproof_vest: 33.75 , Microphone: 33.12 , Traffic_Light: 32.50 , Carton: 31.25 , Volley_Ball: 30.62 , Vending_machine: 30.62 , Lawn_Mower: 29.38 , Car_Wheel: 29.38 , Harmonica: 28.12 , Lighter: 27.50 , Carousel: 27.34 , Mailbox: 27.19 , Airliner: 27.19 , Butternut_Squash: 26.95 , Sweatshirt: 26.56 , Sock: 25.62 , French_Loaf: 25.00 , Dial_telephone: 24.61 , Rabbit: 24.06 , Remote_controler: 22.81 , Modem: 22.50 , Chain-saw: 21.35 , Screwdriver: 20.31 , Power-drill: 19.69 , Electric_Fan: 19.06 , HairDryer: 18.75 , Purse: 18.12 , Wallet: 17.50 , Sunglasses: 17.50 , Minivan: 17.50 , Cat: 15.94 , Microwave: 15.62 , Candle: 15.62 , Mushroom: 15.31 , Dining_Table: 14.06 , Ice_Cream: 13.75 , Perfume: 13.44 , Komodo_dragon: 13.44 , Bycicle: 13.12 , Wooden_Spoon: 12.81 , JellyFish: 12.81 , Canoe: 12.81 , Radio: 12.19 , Desk: 12.19 , African_crocodile: 11.88 , Hatchet: 11.25 , Sandal: 10.00 , Stonewall: 9.69 , Burrito: 9.38 , Palace: 9.06 , Mouse: 7.50 , Convertible: 7.19 , Espresso_maker: 6.88 , Can_Opener: 6.56 , Jeep: 6.25 , Fox: 6.25 , Tile_Roof: 5.86 , Street_Sign: 5.62 , WarPlane: 5.47 , Frog: 5.47 , Wolf: 5.31 , Whale: 5.00 , Torch: 5.00 , Soup_Bowl: 4.69 , Great_white_shark: 4.38 , Kangarou: 3.91 , Digital_Watch: 2.81 , Skirt: 2.50 , Computer_Keyboard: 2.50 , Piano: 1.88 , Manhole_Cover: 1.88 , Bridge: 0.94 , Aircraft_Carrier: 0.39 , Screen: 0.31 , Locomotive: 0.31 , Submarine: 0.00 , Shirt: 0.00 , Loupe: 0.00

## C.4 Additional PUG:SPAR experiments

Instead of using all the background environments presented in the main part of the paper, we also introduce a much simple setup in which we have a single background (thus we do not need the background information in the caption anymore). The background that we choose is the simplest one named "salt flats". This is also the background for which the retrieval accuracy is the highest across all the backgrounds. In table 11, we present the performances of several VLMs when using this single environment. We can observe a significant boost in accuracy for the single object detection task for which the best model achieve 94% accuracy (this value is to contrast with the 78% accuracy obtained across all the background). This show that VLMs are definitively not robust to background changes. However as in the previous case, when probing for the positional information, the performance of the model is still decreasing significantly. We also illustrate in Figure 12 very simple failure cases on the best model.

| Caption | Texture | ViT-B-32 (OpenAI CLIP) | ViT-B-32 (OpenClip 2B) | ViT-L-14 (OpenAI CLIP) | ViT-L-14 (OpenClip 2B) | ViT-H-14 (OpenCLIP 2B) | ViT-G-14 (OpenCLIP 2B) | Flava | BLIP | XVLM | NegCLIP |
|---|---|---|---|---|---|---|---|---|---|---|---|
| *"A photo of a [character]"* | Default | 57.81 | 75.78 | 91.41 | 88.28 | 94.53 | **94.53** | 64.06 | 78.91 | 67.19 | 64.84 |
| | Blue/Red | 48.44 | 54.69 | 71.88 | 70.31 | 75.00 | **84.38** | 42.19 | 53.12 | 48.44 | 48.44 |
| | Grass/Stone | 39.06 | 45.31 | 67.19 | 68.75 | 76.56 | **78.12** | 39.06 | 56.25 | 45.31 | 50.00 |
| *"A photo of a [character] on the (left/right) of the picture"* | Default | 29.69 | 37.50 | 39.06 | 40.62 | 50.00 | **51.56** | 32.81 | 42.19 | 34.38 | 31.25 |
| *"A photo of a [character] on the (bottom/top) of the picture"* | Default | 29.69 | 23.44 | 45.31 | 43.75 | **46.88** | 45.31 | 23.44 | 34.38 | 34.38 | 23.44 |
| *"A photo of a [character] and a [character]"* | Default | 24.56 | 32.47 | 68.85 | 59.67 | 72.66 | **80.96** | 18.55 | 40.58 | 24.71 | 21.04 |
| | Blue/Red | 10.84 | 18.55 | 38.57 | 30.66 | 34.28 | **51.56** | 9.86 | 10.94 | 3.12 | 8.01 |
| | Grass/Stone | 9.38 | 18.16 | 31.35 | 30.47 | 30.18 | **47.85** | 6.84 | 11.33 | 5.96 | 8.98 |
| *"A photo of a [character] on the left and a [character] on the right"* | Default | 14.11 | 14.92 | 38.10 | 30.85 | 40.83 | **42.14** | 13.71 | 23.99 | 14.01 | 14.21 |
| *"A photo of a [character] on the bottom and a [character] on the top"* | Default | 12.00 | 15.42 | 34.07 | 35.48 | 44.05 | **45.26** | 10.99 | 26.51 | 14.62 | 8.17 |
| *"A photo of a [character] textured with [texture1] and a [character] textured with [texture2]"* | Blue/Red | 4.44 | 6.65 | 19.15 | 15.52 | 20.56 | **29.84** | 6.15 | 10.28 | 5.44 | 4.13 |
| | Grass/Stone | 3.43 | 5.44 | 15.73 | 17.24 | 19.05 | **27.32** | 5.54 | 7.76 | 6.35 | 4.03 |

Table 11: Setup and zero-shot evaluation of CLIP models on PUG: SPAR with **caption retrieval in a single environment**. In contrast with the figure presented in the main paper, we present the result only using the salt flats environment. The motivation for this experiment is to showcase the failures mode of VLMs in a very simple setup in which the model robustness to background does not impact the prediction.

## C.5 CLIP fine-tuning details

We utilize the OpenCLIP framework [Ilharco et al., 2021] for all our CLIP experiments. The ViT-B/32 model is used as the image encoder for all our experiments. The CLIP model we fine-tune is the OpenAI 400M pre-trained model *('ViT-B-32', 'openai')*[10]. Fine-tuning on PUG: AR4T is done for 10 epochs on the 200K dataset while training is done for 2 epochs on the 1M dataset.

---

[10]NOTE: We do not perform any training on the proprietary 400M dataset from OpenAI. We strictly only use the pre-trained models released, and fine-tune them on our PUG datasets.

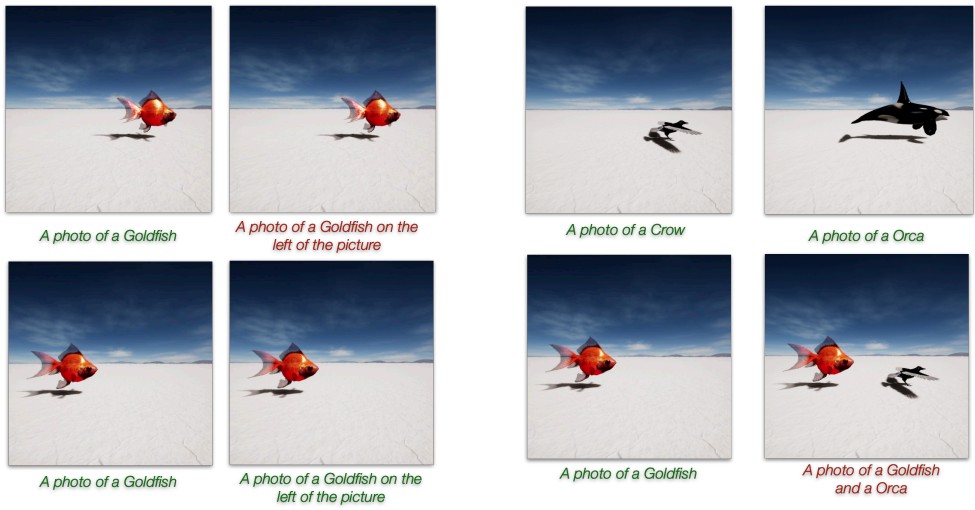

Figure 12: Failures mode of a OpenCLIP ViT-G-14. Our PUG: SPAR dataset provides very simple images and captions and yet even large models are failing on them.

