# OpenReview forum: "PUG: Photorealistic and Semantically Controllable Synthetic Data for Representation Learning"
_NeurIPS.cc/2023/Track/Datasets_and_Benchmarks — NeurIPS 2023 Datasets and Benchmarks Poster_

### Official Review · Reviewer_bwEW · 2023-07-03
**Reviewer bwEW**

**Rating:** 7
**Confidence:** 3

**Strengths:**

* The paper proposes a semantically controllable framework for rendering 2D images from a 3D scene using the Unreal engine. Such research direction is valuable as it enables to evaluate computer vision models in controlled scenarios, which would not be feasible with with real natural images. As such it is valuable that the paper explores factors of variation related to the camera pose, the object pose, the object size, the object texture, the lighting of the scene, or the scene itself.
* As mentioned in the paper, synthetic data also have the advantage of presenting less issues related to privacy, bias and copyrights than real natural images.
* The paper proposes relevant benchmarks that rely on the rendered images. It showcases the potential of such images in the computer vision context as it helps to highlight issues related to the model robustness or how such models can be fine-tuned for improved performance.

**Additional Feedback:**

* L196-206: I believe the text should refer to figure 14 rather than figure 9.
* There is a mismatch between the main paper and the appendix regarding PUG: ImageNet as numbers differ

**Clarity:**

* The paper is well-written and easy to follow.
* Figures and tables are in general clear and informative.
  * Figure 4 (right) would deserve better y-axes. In the current form, it is very hard to distinguish the difference among the different models.

**Correctness:**

* The proposed dataset is constructed in a sound way. The paper uses the Unreal Engine to render the 2D images.
* The proposed benchmarks are appropriate and illustrate the potential of the dataset for evaluating equivariance and robustness about various image factors (e.g., pose, size, texture...), and for fine-tuning models on a compositionality task.

**Documentation:**

* **[Code]** The paper promises to release a python API, TorchMultiverse. This is currently not available.
* **[Rendering assets]** To my understanding, the authors will not release the 3D objects and environments. To reproduce the rendering steps, the community will have to acquire these, and follow the steps described in the appendix. It is unclear how easy these steps are.
* **[Licenses of existing assets]** The appendix lists the licenses for the 3D objects and environments used to render the dataset. It is appreciated that the authors provided detailed information about these assets.
* **[Image dataset]** The paper will release the rendered images. After a quick look, they seem fine. At the moment, there is no way to verify how the dataset will be hosted, licensed or maintained though.
* **[Image models]** Models used for benchmarking will not be released. This might be problematic for the adoption of the dataset as some benchmarks might be quite tricky to reproduce as the information is a bit limited on how to fine-tune models (e.g., Table 3).

**Ethics:**

None.

**Limitations:**

* Limitations are discussed in the appendix and are sufficient.

**Opportunities For Improvement:**

* The paper lacks a characterization of photorealism
  * The paper claims to offer a higher degree of photorealism than existing datasets in the field. Such claim has not been assessed, as we do not know to what degree the proposed framework is better in terms of photorealism than the previous ones.
  * The paper proposes several benchmarks to evaluate existing computer vision models. For example, in Table 1, several models are evaluated to understand their performance w.r.t. several factors of variation. It appears that all models exhibit a drop in performance when evaluated on the rendered dataset when compared with the original performance on the ImageNet validation set. How much is this drop attributed to the domain gap vs the variation change?

* The parameters set for rendering lack motivation and consistency
  * The number of images rendered per dataset seems quite arbitrary and lacks motivation. The animal set has 365k images, the ImageNet set has 90k images, and the Attribute set has 1M+ images. How were those number of images chosen? and more generally, what would be the minimum of images to render for each task to obtain significant and reliable results?
  * The different proposed datasets have different scenes or parameters. For example when comparing the Animal and ImageNet sets, why is the number of backgrounds, sizes or textures different? Why not use a similar number?

* The rendering process lacks details about object positioning
  * There is no discussion in the paper about where objects are positioned in the image, and whether it has an influence over the final results. For example, it would have been interesting to have a saliency map to know whether objects mostly appear in the center of the image.
  * There is no discussion in the paper about object affordance. How do the objects interact with the scene? Is it realistic and does it have an influence on the results? For example in some images in Figure 5 or 6, it seems like the objects are floating in the air.

* The rendering process lacks details about compute
  * What is the compute needed to render 2D images? and how long does it take?

**Relation To Prior Work:**

* The related work is sufficient to cover the use of rendered data for representation learning.
* It would have been interesting to have a discussion about generative models for representation learning too. Some examples include:
  * Li et al. BigDatasetGAN: Synthesizing ImageNet with Pixel-wise Annotations. CVPR 2022
  * He et al. Is synthetic data from generative models ready for image recognition? ICLR 2023
  * Sariyildiz et al. Fake it till you make it: Learning transferable representations from synthetic ImageNet clones. CVPR 2023
* The experiment on OOD with *PUG: Animals* lacks grounding with existing works, so it is hard to relate to which line of works the authors refer to when discussing about OOD.

**Summary And Contributions:**

The paper addresses the relevance of rendered images for evaluating existing computer vision models or fine-tuning vision-language models. To achieve this, the authors propose to rely on the Unreal engine to render photorealistic images. The main benefits lie in the controllability of the rendered images, as it becomes possible to control for factors related to camera pose, object pose, object size, scene lighting, and so on, which is useful and pertinent for model evaluation. As such, the authors propose three main use cases: equivariance of the feature space with images of animals, robustness of the prediction with images of ImageNet classes, and compositionality in vision-language models with images of two objects in a scene.

---

> ### Author Response · Authors · 2023-08-17
>
> Thank you very much for your very detailed review and all the very insightful comments ! Please take a look at our general answer and also at the website and codebase.
>
> > The paper claims to offer a higher degree of photorealism than existing datasets in the field. Such claim has not been assessed, as we do not know to what degree the proposed framework is better in terms of photorealism than the previous ones.
>
> In our work, we consider photorealism as producing images whose characteristics(quality/diversity/richness) are close to natural images (with realistic lightning or shadow). There is also a very wide range of different datasets as we mentioned in the related work which could be close to photorealism. For example some indoor dataset for reinforcement learning have really great graphics, however their scope is too narrow. There is the biased car dataset (https://github.com/Spandan-Madan/generalization_to_OOD_category_viewpoint_combinations) which shows realistic images however they have only a city road environment. In that case, it is not just that the images might be higher quality than the ones in Biased-car (which might be subjective). It is also that we offer a diversity in terms of environment that was not available in this dataset. As suggested by a reviewer, we added Table 5 in the appendix that compares the characteristics of our PUG: Animal dataset with other datasets such as Biased-Car.
>
> In addition, what we are introducing is also a framework which is only bound by the latest capabilities of the most powerful video game engine that is available on the market. In this work, we used Unreal Engine 5.0, however the version 5.2 offers much more photorealistic rendering of foliage which could be leveraged for new PUG datasets. This is mostly what motivated our claim about offering a higher degree of photorealism than what was done before since all other approaches have used less powerful (with lower quality) rendering engines.
>
> > The paper proposes several benchmarks to evaluate existing computer vision models. For example, in Table 1, several models are evaluated to understand their performance w.r.t. several factors of variation. It appears that all models exhibit a drop in performance when evaluated on the rendered dataset when compared with the original performance on the ImageNet validation set. How much is this drop attributed to the domain gap vs the variation change?
>
> This is a very good question. To address it, we display the accuracy per class on our PUG:ImageNet benchmark in the appendix (page 33). There are only 3 classes for which the network is never able to predict the classes. However for all the other classes, there is always a specific combination of factors for which the network is successful in being able to predict the correct class (in that case, the network is probably failing because of the combination of factors instead of the gap sim-real). If users are concerned about the sim-to-real gap, they can create a subset of the PUG: ImageNet dataset which only contains the classes for which the accuracy is higher than a given threshold.
> If we look at PUG: SPAR, we have included zero-shot results for single animal detection only. As shown in the appendix, in the best environment, the best model achieves 94% in accuracy meaning that the sim-to-real gap is not an issue in this case.
>
> > The number of images rendered per dataset seems quite arbitrary and lacks motivation. The animal set has 365k images, the ImageNet set has 90k images, and the Attribute set has 1M+ images. How were those number of images chosen? and more generally, what would be the minimum of images to render for each task to obtain significant and reliable results?
>
> The number of images for each dataset is based on ensuring a rich coverage of the number of factors needed for each. For PUG: Animals, it is mostly because we have all possible combinations of the factors of variations. For example we have 64 backgrounds, however adding more backgrounds would have been more costly. So we thought it was better to have 64 of them, than trying to round up to something like 50. So to answer this question, this is mostly a limitation with respect to the number of assets we have. In addition to PUG: Animal, we also wanted to have enough data to be able to train on this dataset and to have a big enough evaluation/test split. Concerning PUG: ImageNet, since this dataset is not for training, we could significantly reduce the size of the dataset. And because this is for evaluation purposes, we do not want the evaluation to take too long. In addition several academic labs have limitations in terms of compute/stockage, so releasing an evaluation dataset with 500k images would not contribute in making this dataset a popular benchmark in the field.

---

> > ### Author Response · Authors · 2023-08-17
> >
> > > The different proposed datasets have different scenes or parameters. For example when comparing the Animal and ImageNet sets, why is the number of backgrounds, sizes or textures different? Why not use a similar number?
> >
> > In the revision of the paper and datasets, we now have the same number of environments for PUG: Animal and ImageNet. Concerning the differences in the number of size, textures, camera variations, it is mostly because for PUG: Animal we wanted to have all the combinations of factors of variations. In that case, if we had used the same number of factors as we did for PUG: ImageNet, we would have ended up with 14M images in PUG: Animal ! We think that it would be unlikely for most people to use such a big dataset, so we decided to restrict it to a fewer number of factors.
> >
> > > There is no discussion in the paper about where objects are positioned in the image, and whether it has an influence over the final results. For example, it would have been interesting to have a saliency map to know whether objects mostly appear in the center of the image.
> >
> > There is no need for a saliency map since we have a controllable environment in which we decide where the objects are placed. All objects are placed in the center of the images (In the exception of the PUG: SPAR and PUG:AR4T datasets).
> >
> > > There is no discussion in the paper about object affordance. How do the objects interact with the scene? Is it realistic and does it have an influence on the results? For example in some images in Figure 5 or 6, it seems like the objects are floating in the air.
> >
> > The objects are placed in the 3D scene so it is realistic (since we have all the corresponding shadows and reflections). For PUG: Animal and PUG: SPAR the animals are never in the air. However for PUG: ImageNet, it can happen that some of the assets might end up in the air (since they are coming from different sources and loaded at runtime, they do not always have the correct offsets).
> >
> > > The rendering process lacks details about compute
> >
> > It takes around 1s to generate a 512x512 image. We parallelize the sampling across our 64 environments. In that instance, each gpu samples the images associated with a specific background (since switching between the 3D scene is the operation that takes the longest time). For a dataset like PUG: Animals, which contains around 200K images, we parallelize the sampling across 64 GPUs. It took around 1h to sample this dataset. It would have taken a bit more than 2 days using a single gpu. We updated the paper accordingly to add this information in the main text.
> >
> > > Figure 4 (right) would deserve better y-axes. In the current form, it is very hard to distinguish the difference among the different models.
> >
> > Thanks, we have updated the draft !
> >
> > > It would have been interesting to have a discussion about generative models for representation learning too
> >
> > Thank you for the suggestion, we have added a paragraph on this in “Sec 2. Related Works” in our revised manuscript!
> >
> > > The experiment on OOD with PUG: Animals lacks grounding with existing works, so it is hard to relate to which line of works the authors refer to when discussing about OOD.
> >
> > Thank you for the suggestion, we update the PUG: Animals section with more related work around OOD. We also added a table as suggested by a reviewer in the appendix which compares PUG: Animals to other OOD datasets.
> >
> > > Additional Feedback
> >
> > Thanks, we updated the paper accordingly !

---

> > > ### Comment · Reviewer_bwEW · 2023-08-21
> > >
> > > Thank you for the rebuttal. It addresses most of the points I have raised in the review. The release of the website also helps in clarifying several other questions.

---

### Official Review · Reviewer_b1EE · 2023-07-17
**Review of Submission 820**

**Rating:** 5
**Confidence:** 4
**Clarity:** The paper is well written and easy to…

**Strengths:**

The controllability of the dataset is a major strength of PUG. The idea of using game engines for representation learning research is interesting and promising.

**Additional Feedback:**

Please see "Opportunities For Improvement".

**Correctness:**

The claims are correct and the dataset is synthetically constructed in a sound way.

**Documentation:**

There is sufficient detail on data collection and organization.

**Ethics:**

I found no ethical problems.

**Limitations:**

The authors have adequately addressed the limitations.

**Opportunities For Improvement:**

I like this paper and enjoyed reading it, but I think the benchmarking settings and experiments need to be more refined. Below are my concerns and questions.

(1) Lack of comparison with the existing OOD benchmarking methods.

PUG:Animals provides a new and interesting way of OOD benchmarking. However, it is unclear why the proposed benchmarking is important and how it differs from other OOD benchmarking methods. I think a narrower focus and discussion of the importance of measuring OOD performance on synthetic data is needed. For example, the following paper focuses on the class OOD and discusses various detailed levels of detection difficulty with many models including ResNet, EfficientNet, MLP Mixer, ViT, etc.

[A] Ido Galil, Mohammed Dabbah, and Ran El-Yaniv, A framework for benchmarking Class-out-of-distribution detection and its application to ImageNet, ICLR, 2023.

(2) Pre-training performance compared to ImageNet.

In Section 3.4, PUG:Attributes&Relations is used for pre-training vision-language models. If the goal is to pre-train models, I would like to see the performance of vision-only models compared to pre-training on ImageNet. I also think a comparison with Shaders21k [B], which uses OpenGL for representation learning, would be interesting.

[B] Manel Baradad, Chun-Fu Chen, Jonas Wulff, Tongzhou Wang, Rogerio Feris, Antonio Torralba, and Phillip Isola, Procedural Image Programs for Representation Learning, NeurIPS, 2022.


**Relation To Prior Work:**

Yes, the authors clearly discussed how this work differs from previous works.

**Summary And Contributions:**

This paper provides PUG, a 3D graphics environment for rendering image data for representation learning. PUG uses the Unreal Engine (a game engine developed by Epic Games) to generate synthetic datasets for benchmarking the robustness and generalization ability of image recognition models such as ResNet, ViT, and CLIP. Specifically, three datasets are presented: PUG:Animals, PUG:ImageNet, and PUG:Attributes&Relations. PUG:Animals consists of 365k images of 68 animals with 56 backgrounds and is used to evaluate out-of-distribution recognition performance. PUG:ImageNet consists of 90k images of 755 assets representing 151 ImageNet classes with 62 backgrounds. This dataset is used to evaluate robustness. PUG:Attributes&Relations contains image-caption pairs for training CLIP (vision language models).

---

> ### Author Response · Authors · 2023-08-17
>
> Thank you for your review. We invite you to look at the general answers as well as taking a look at the website and codebase.
>
> > Lack of comparison with the existing OOD benchmarking methods
>
> Thanks for pointing that out, we updated the pdf with a better discussion and related work around the OOD benchmarking. To answer your comment, the paper you are sharing focuses on how robust pretrained models on ImageNet can be with respect to different factors. In our paper, this is the focus of PUG: ImageNet and not PUG: Animals. PUG: Animals is not aimed to measure the OOD robustness of pre-trained classifier (in fact many of the animals in PUG: Animals are not even in the list of ImageNet classes). PUG: Animals is meant for studying the dynamics of generalization to unseen factors (or unseen combinations of factors) during training. In that instance, a user will train from scratch a model on PUG: Animal and evaluate on the unseen factors how the model generalizes. This is closer to what can be done with the Nico++ dataset or biased cars dataset.
>
> > Pre-training performance compared to ImageNet
>
> As explained in the general comment, the ARO benchmark that we used in the first version of the paper was shown to be unreliable and that the performances on our PUG: SPAR benchmark did not demonstrate statistically significant improvement with respect to the CLIP model. So we decided to de-emphasize the part about fine-tuning focusing this paper on evaluations with the introduction of PUG: AR4T which can be used for fine-tuning at the end. We still include the performance on ARO, but we do not claim anymore that training on synthetic data is improving the performance of VLMs. This will be the subject of a follow-up paper since as you noted, training on synthetic data will require much more comparisons to the existing literature around this topic (as well as an ablation with respect to the number of images and diversity…).

---

### Official Review · Reviewer_7VBW · 2023-07-20
**Controllable Photorealistic Synthetic Data Generation**

**Rating:** 8
**Confidence:** 4
**Correctness:** Approach is technically sound
**Clarity:** Yes

**Strengths:**

1. Idea of PUG is clear and well-motivated.
2. Multiple use cases have been discussed and evaluated
3. Paper is well-written and can be widely-used by the community

**Additional Feedback:**

I hope the code/web-site/documentation will be available for a wider audience as soon as possible.

**Documentation:**

Data examples are available, code/web-site/documentation is not yet available.

**Limitations:**

Code/web-site are not published yet, only the data examples.

**Opportunities For Improvement:**

Once the code is released, it is important to have a good documentation how to setup environment and controllably generate data of interest. I hope, the code will be general enough to be extended for all kinds of objects in the data. Also, it is important to optimize the complexity of data generation.

**Relation To Prior Work:**

Yes

**Summary And Contributions:**

This paper proposes an environment for controllable photorealistic synthetic data generation (PUG) using Unreal Engine. This approach shows effectiveness for many purposes: model pretraining (including VLM), study of generalization to OOD etc. PUG can be a complementary and effective option to non-synthetically trained VLMs but with lower complexity.

---

> ### Author Response · Authors · 2023-08-17
>
> Thank you for your review and comments ! We invite you to read the general comment and take a look at the website and the Github repository.
>
> > Once the code is released, it is important to have a good documentation how to setup environment and controllably generate data of interest. I hope, the code will be general enough to be extended for all kinds of objects in the data. Also, it is important to optimize the complexity of data generation.
>
> We agree good documentation is important for adoption across the research community. Please let us know if you have any comment with respect to the codebase on Github. We also added jupyter notebooks on the Github to enable dataset visualization and make it easier to use. Concerning using all kind of objects, it is already possible with the Unreal plugin https://github.com/rdeioris/glTFRuntime which loads .glb files at runtime (we used it to create PUG: ImageNet and PUG: AR4T).

---

> > ### Comment · Reviewer_7VBW · 2023-08-21
> >
> > Thanks for releasing your code and data!

---

### Official Review · Reviewer_hTqw · 2023-07-21
**Detailed construction method and good evaluation**

**Rating:** 6
**Confidence:** 4
**Correctness:** Yes
**Clarity:** Yes

**Strengths:**

1.	The process of how to construct the dataset is detailly described.
2.	The paper is easy to read and follow.
3.	The targeted problem is worth exploring.


**Additional Feedback:**

No additional feedback

**Documentation:**

It lacks a URL for the reviewer access to the dataset.

**Ethics:**

No concerns.

**Limitations:**

The authors have addressed the limitations.

**Opportunities For Improvement:**

1.	To be compared with existing datasets with the same usage, a table can be useful. For example, image numbers and the number of object classes can offer valuable information about the progress achieved.
2.	The experiments in the paper show that training on synthetic data could improve the performance of CLIP models on real data. A figure or table to show the accuracy improvement with data increase will be better.

**Relation To Prior Work:**

Yes

**Summary And Contributions:**

This paper uses Unreal Engine to produce many fake images and build a PUG dataset. Robust evaluation and model training are conducted in this dataset.

---

> ### Author Response · Authors · 2023-08-17
>
> Thank you for your review. We updated the paper and also added the URL towards the website and Github repository.
>
> > To be compared with existing datasets with the same usage, a table can be useful. For example, image numbers and the number of object classes can offer valuable information about the progress achieved.
>
> This is a good point, we added such a comparison in Table 5 in the appendix.
>
> > The experiments in the paper show that training on synthetic data could improve the performance of CLIP models on real data. A figure or table to show the accuracy improvement with data increase will be better.
>
> In our original submission, we had noted in Table 3 the utility of using PUG for fine-tuning to improve performance on the Attributes, Relations, and Order (ARO) benchmark which is a real data benchmark made from Visual Genome, Flickr-30K, and MS COCO Captions.
> In the revised main paper, we have a comparison between fine-tuning with 200K and 1M PUG samples. While we observed an increase in performance on the ARO dataset, recent works have shown that ARO is not very reliable as a benchmark. As mentioned in the common rebuttal, this is why we introduced the PUG:SPAR benchmark to test compositionality in VLM models. . Since the main contribution of this work is to provide new datasets (and we are already at the 10 page limit), we did not pursue the fine-tuning experiments further. We believe that solving relation understanding in VLMs should be a work on its own.  Hence we release the dataset for the research community to explore this interesting question.

---

### Official Review · Reviewer_PGeG · 2023-07-24
**All around good work; a useful new platform and resource**

**Rating:** 7
**Confidence:** 3
**Correctness:** I did not find any significant errors…

**Strengths:**

* The data appears to be more photorealistic than prior large-scale 3D datasets
* The datasets are large-scale; significant effort went into making this
* Evaluation covers a good range of recent and popular vision foundation models
* Unlike real datasets, the synthetic data can be manipulated more systematically to ask more pinpointed questions about model robustness and generality
* Good to see that training on the data can substantially increase CLIP's performance, without many tricks

**Additional Feedback:**

None.

**Clarity:**

The paper is overall clear, if a bit dense. I would suggest the following improvements:

1. It would be useful to write out the equations for measuring equivariance in the main text, and to provide a clearer figure showing the geometry of the embeddings alongside notation for how equivariance is measured.
2. For the OOD experiments, it could help to have a schematic plot showing which factors are varied and what parts of the full space of combinations is sampled from for both train and test. It could be a bit like Figure 1 in https://arxiv.org/abs/1810.12282
3. Table 1: It takes some thought to understand what these numbers represent. I think they refer to top-1 accuracy on the set of images where each factor is varied. It would help to spell this out more concretely, perhaps with an equation for the metric.
4. Fig 3: "Training Size" could be read as number of training examples; perhaps "Asset Size"
5. "object-attribute associations ('the paved road and the white house' vs 'the white house and the paved road')" -- Is there a typo in this example? I don't understand the difference.

**Documentation:**

There is fairly detailed documentation in the supplemental material but I would really like to see a project webpage and code. The authors promise this is coming. Especially this will be important for the tutorial on using the Unreal Engine. The pdf tutorial in supplemental section B is short and I think it will be hard to follow. Step by step code on GitHub or another website would likely be preferred.

**Ethics:**

I do not have concerns.

**Limitations:**

The limitations section (A.1) is short and barely talks about concrete limitations, instead just saying that more could be done. This section would add a lot more if it were more precise. What about limitations in terms of 1) difficulty of creating assets, 2) runtime (mentioned a bit w.r.t. RL), 3) lack of diversity in the data / biases, 4) specific sim2real gaps that remain, and their effects on the paper's experiments, and so on. A richer discussion would be appreciated.

**Opportunities For Improvement:**

* The paper is quite dense and clarity can be improved in a few place (see clarity section below).
* There are many existing synthetic datasets. The current paper does a good job arguing why these have limitations, but there are few concrete comparisons. I appreciated the comparison to Syn-CLIP, which is trained on a different synthetic dataset. However, I'm still left wondering if some of the same things as shown in this paper could be just as well shown using other datasets. I don't see it as critical, but it could strengthen the contribution to show more head-to-head comparisons where the same experiments are run using prior datasets versus using PUG.
* No discussion of synthetic data from generative models (see "relation to prior work" section below).

**Relation To Prior Work:**

I think the prior work section on 3D synthetic data (computer graphics engines) is adequate. I don't think it is nearly comprehensive but it touched on the main competitors, which I think is sufficient for a 9 page paper.

However, one area I found missing is a discussion of synthetic data from generative models (GANs, diffusion models, etc). I think it is worth mentioning this parallel approach since there are many recent papers on using generative models to create datasets for probing and training visual representations. These approaches could be seen as direct alternatives to the proposed platform. Some representative papers are listed below:
* Sariyildiz, Mert Bulent, et al. "Fake it till you make it: Learning (s) from a synthetic ImageNet clone." CVPR (2023).
* Jain, Saachi, et al. "Distilling model failures as directions in latent space." ICLR (2023).
* He, Ruifei, et al. "Is synthetic data from generative models ready for image recognition?." ICLR (2023).
* Jahanian, Ali, et al. "Generative models as a data source for multiview representation learning." ICLR (2022).
* Zhang, Yuxuan, et al. "DatasetGAN: Efficient labeled data factory with minimal human effort." CVPR (2021).


**Summary And Contributions:**

This paper introduces a new platform (PUG), and accompanying datasets, for 3D synthetic data. The main selling point relative to prior synthetic datasets is the level of photorealism, which is achieved by using the Unreal Engine. Three datasets are presented using this platform. These demonstrate the ability of synthetic data to probe model robustness and generality through systematic variation of different causal factors (lighting, material, etc.). These datasets are used to evaluate existing vision foundation models and also to finetune one model (CLIP) to achieve better performance on attribute binding and identifying inter-object relationships.

---

> ### Author Response · Authors · 2023-08-17
>
> Thanks a lot for your review and positive comments !
>
> > There are many existing synthetic datasets. The current paper does a good job arguing why these have limitations, but there are few concrete comparisons. I appreciated the comparison to Syn-CLIP, which is trained on a different synthetic dataset. However, I'm still left wondering if some of the same things as shown in this paper could be just as well shown using other datasets. I don't see it as critical, but it could strengthen the contribution to show more head-to-head comparisons where the same experiments are run using prior datasets versus using PUG.
>
> You are right that there are many existing synthetic datasets out there for which it is probably possible to show several of the things we demonstrated with PUG. For example in [1], the authors demonstrate the lack of robustness of pre-trained models in a similar way as we did with a synthetic dataset. They evaluated robustness to pose changes where they observed a consistent drop in SoTA model performance. However, they were not able to release their synthetic datasets while we were able to show similar results while releasing the data associated with these experiments.
>
> There are also several works like biased card [2] which also use synthetic datasets to study OOD generalization, however it contains 30K images (whereas our PUG: Animals contain 200K images). To get a better understanding of the differences with other datasets, we added Table 5 in the appendix. We hope that this will solve your concern. Even if it is possible to show some of the behaviors we demonstrated with other smaller and lower quality synthetic dataset, we think that researchers will probably prefer to use higher quality and larger scale  synthetic datasets as the one we introduce.
>
> [1] The Robustness Limits of SoTA Vision Models to Natural Variation, Ibrahim et al 2022,
>
> [2] When and How CNNs generalize to out-of-distribution category-viewpoint combinations.Madan et al 2021
>
> > No discussion of synthetic data from generative models
>
> We agree such a comparison is important, we updated the paper with the discussion around generative models. Thank you for the references.
>
> > It would be useful to write out the equations for measuring equivariance in the main text, and to provide a clearer figure showing the geometry of the embeddings alongside notation for how equivariance is measured
>
> We update the paper accordingly, thank you for pointing this out.
>
> > For the OOD experiments, it could help to have a schematic plot showing which factors are varied and what parts of the full space of combinations is sampled from for both train and test
>
> Concerning the asset size and the textures,  we train on the specific value that can be read on the x axis while evaluating all the factors which do not have such value. So if we train on the size medium, we evaluate on small and large. If we train on the Default texture, we evaluate on Grass/Sky/Asphalt. Concerning the backgrounds, it is the same, if we train on 50 backgrounds, then the evaluations are done with (64: total number of backgrounds - 50) 14 backgrounds. If we train on 10 backgrounds, we evaluate on 54 backgrounds. So, the training factors are already shown in the figure, to get the test factors, we only need to get the number of total factors minus the number of train factors. We updated the figure’s caption for this to be more clear.
>
> > “Table 1: It takes some thought to understand what these numbers represent. I think they refer to top-1 accuracy on the set of images where each factor is varied. It would help to spell this out more concretely, perhaps with an equation for the metric.”
>
> Yes, they are all the top-1 accuracy on the given subset!  We have added the description at the top of the table so readers don’t miss it.
>
> > “Fig 3: "Training Size" could be read as number of training examples; perhaps "Asset Size"”
>
> Thanks, we updated the figure !
>
> > object-attribute associations ('the paved road and the white house' vs 'the white house and the paved road')" -- Is there a typo in this example? I don't understand the difference
>
> Thanks for noting the typo. In our updated submission, we removed the problematic text..
>
> > “There is fairly detailed documentation in the supplemental material but I would really like to see a project webpage and code. The authors promise this is coming. Especially this will be important for the tutorial on using the Unreal Engine. The pdf tutorial in supplemental section B is short and I think it will be hard to follow. Step by step code on GitHub or another website would likely be preferred.”
>
> This is a good point, GitHub will make it easier for researchers to build on our work. We moved the tutorial on GitHub since it will be easier to get feedback and to update it if needed. Please look at our general answer which contains the Github and website link !

---

> > ### Comment · Reviewer_PGeG · 2023-08-20
> > **thanks for the updates**
> >
> > Thanks for updating the paper to address my concerns! I think it's a solid submission and should be accepted.
> >
> > I'm satisfied with the discussion around prior datasets, and table 5 is nice to see, but I also think it can still be interesting to add more direct comparisons where you show that PUG evaluates OOD performance more meaningfully than the prior datasets do. I'm not quite sure how to design such a comparison but one idea would be to measure consistency across multiple different kinds of OOD tests. Perhaps PUG more holistically evalutes OOD settings, so that a held out OOD setting can be better predicted.

---

### Author Response · Authors · 2023-08-17

We would like to thank all the reviewers for their time and very insightful comments. We are glad to see such appreciation for our PUG datasets and we are very grateful to the reviewers for highlighting how promising such an approach is.

We recently released publicly our PUG: datasets with a website that you can see there:
https://pug.metademolab.com/

While the code is available there (with the list of asset, tutorials and jupyter notebooks):
https://github.com/facebookresearch/PUG

We also updated the paper following the reviewer’s suggestions. We invite you to look at the last revision we submitted. Thank you again for helping us improve our work.

List of changes:
- Split the PUG: Relation and Attributes dataset into two distinct datasets: PUG: SPAR (Spatial, Position, Attribute and Relation) which should be used to evaluate VLMs and PUG: AR4T which is the set that is used for fine-tuning VLMs. We carefully build these two distinct datasets such that their assets do not overlap. We decided on this change because the very recent work of [1] (that appears at the same time of our submission) highlighted the unreliability of the VLMs benchmarks such as ARO. In the absence of a reliable benchmark for VLMs, we decided to leverage our PUG framework to introduce a new and reliable benchmark for VLMs named PUG: SPAR. In consequence, we added a new section in the paper that evaluated state of the art VLMS on the PUG: SPAR dataset. We also deemphasize the fine-tuning results since the improvement we got was on the ARO benchmark which is now known to be unreliable (which was not the case at the time of publication).
- Added the related work on synthetic data with generative models as suggested by the reviewers. Thank you again for the references !
- Move the Unreal Engine tutorial on the github (which is a better support for this kind of content as highlighted by some reviewers). This also makes it easier for the community to raise questions about the setup and to use our framework.
- Move also the assets list into the github (as well as an additional file in each of the compressed dataset file)
- Move the OOD experiments into the appendix and better motivate the use of PUG: Animals in an OOD scenario in the main paper.
- Updated the equivariance section with the formula and better labels for the figure.
- Added Table 5 in the appendix to compare PUG: Animals with other datasets
- Added information about the compute in the main text

[1] VisualGPTScore: Visio-Linguistic Reasoning with Multimodal Generative Pre-Training Scores, Lin et al. (June 2023)

---

### Decision · Program_Chairs · 2023-09-22

**Decision:**

Accept (Poster)

**Comment:**

The ratings of this paper are 7-6-8-5-7. Reviewers are generally positive. This paper introduces a platform PUG for generating 3D synthetic data using the Unreal Engine. This platform is accompanied by three datasets designed for various purposes, including model evaluation and fine-tuning. Reviewers acknowedge that it is impressive to achieve a high level of photorealism in the synthetic data, surpassing prior large-scale 3D datasets. Besides, the paper effectively covers a range of recent and popular vision foundation models, providing valuable insights into their performance with synthetic data. This paper also demonstrates that training on the synthetic data can significantly enhance CLIP's performance without resorting to complex tricks. Authors addressed reviewers' concerns well in the response. Given these considerations, AC recommends to accept this paper.